# Discovery of a CNS active GSK3 degrader using orthogonally reactive linker screening

Andreas Holmqvist [1,6], Nur Mehpare Kocaturk [1,6], Christina Duncan [2], Jennifer Riley[2], Steven Baginski [2], Graham Marsh [3], Joel Cresser-Brown[3], Hannah Maple[3], Kristiina Juvonen[1], Gajanan Sathe [1], Nicola Morrice[4], Calum Sutherland[5], Kevin D. Read [2] & William Farnaby [1] ✉

Bifunctional targeted protein degraders, also known as Proteolysis Targeting Chimeras (PROTACs), are an emerging drug modality that may offer a new approach for treating neurodegenerative diseases. Identifying chemical starting points for PROTACs remains a largely empirical process and the design rules for identifying Central Nervous System (CNS) active PROTACs have yet to be established. Here we demonstrate a concept of using orthogonally reactive linker reagents, that allow the construction of screening libraries whereby the E3 ligase binder, the target protein binder and the linker can be simultaneously varied and tested directly in cellular assays. This approach enabled the discovery of Glycogen Synthase Kinase 3 (GSK3) PROTACs which are CNS in vivo active in female mice. Our findings provide opportunities to investigate the role of GSK3 paralogs in cellular and in vivo disease models and for the rapid discovery of in vivo quality bifunctional chemical probes for CNS disease concepts.

Proteolysis-targeting chimeras (PROTACs) have emerged as a transformative therapeutic modality with an increasing presence in clinical trials for a spectrum of diseases[1]. These heterobifunctional degrader molecules, composed of an E3 ligase binder, a protein-of-interest (POI) binder, and a linker, induce ternary complexes between the E3 ligase and the POI, catalysing ubiquitination and subsequent degradation of the target protein via the ubiquitin-proteasome system (UPS). This mechanism enables sub-stoichiometric quantities of the drug to induce robust degradation, offering potential advantages over traditional occupancy-driven therapies with respect to efficacy and specificity[2]. Recent advancements have included promising developments targeting the central nervous system (CNS), exemplified by NX-5948, a brain penetrant Bruton's Tyrosine Kinase degrader and ARV-102, a Leucine-Rich Repeat Kinase 2 (LRRK2) degrader indicated for PD[3,4]. Despite these advances, as compared with inhibitors[5], the

property space, design principles and assay cascades required for CNS PROTAC discovery are not established, with a lack of available chemical probes and datasets that could guide further understanding. Given the promise and potential of targeted protein degradation to address hitherto intractable cellular pathways and targets and the clear unmet need and societal challenges associated with diseases of the CNS, this is a striking gap.

PROTAC hit identification remains predominantly empirical and every project requires a bespoke screening library at the outset. Recently, high-throughput 'Direct-to-Biology' (D2B) approaches have made progress in addressing this challenge[6–12]. This has mainly been approached by constructing pre-conjugated E3 ligase binder-linker building blocks and then enabling a single, plate-based, high conversion reaction to attach a protein of interest binder, with testing of crude mixtures directly in cellular degradation assays. Whilst these

[1]Centre for Targeted Protein Degradation, School of Life Sciences, University of Dundee, Dundee, UK. [2]Wellcome Centre for Anti-Infectives Research, Drug Discovery Unit, Division of Biological Chemistry and Drug Discovery, School of Life Sciences, University of Dundee, Dundee, UK. [3]Bio-Techne (Tocris), The Watkins Building, Atlantic Road, Avonmouth, Bristol, UK. [4]Division of Neuroscience, School of Medicine, IMS/WTB Complex, University of Dundee, Dundee, UK. [5]School of Medicine, Ninewells Hospital & Medical School, University of Dundee, Dundee, UK. [6]These authors contributed equally: Andreas Holmqvist, Nur Mehpare Kocaturk. ✉e-mail: w.farnaby@dundee.ac.uk

platforms significantly accelerate PROTAC discovery and structure-activity-relationship exploration, early efforts relied predominantly on amide coupling[7,9,10], resulting in an inevitable influence on the property space of the resulting libraries, for example, by often introducing redundant hydrogen bond donors (HBDs) that can limit cellular or tissue permeability. Recently, towards addressing this gap, Stevens et al. introduced a broader set of medicinal chemistry transformations for D2B PROTAC generation, including reductive amination, $S_N$Ar, palladium-mediated cross-coupling, and alkylation[12]. In addition, Roberts et al. had previously used formation of acyl hydrazones to generate a library of nearly 100 Estrogen Receptor PROTACs[6]. Despite significant progress, the requirement for the synthesis and purification of significant numbers of pre-conjugated intermediates reduces agility when wishing to vary all three components of E3 binder, linker or POI binder in a given library set-up.

Glycogen synthase kinase 3 (GSK3) is a highly conserved serine/threonine kinase with two paralogs, GSK3α and GSK3β, sharing 85% sequence homology and 98% similarity within the kinase domain[13]. Both isoforms have a large number of substrates[14] and regulate several key signalling pathways, including Wnt/β-catenin, insulin and mTOR, which are implicated in cellular functions such as proliferation, differentiation, and apoptosis[15]. In the CNS, GSK3β has been linked to neurodegenerative diseases, contributing to Alzheimer's disease (AD) through tau hyperphosphorylation and amyloid-β production, and to Parkinson's disease (PD) by modulating α-synuclein aggregation and dopaminergic neuron survival[16]. GSK3α, while less studied, has been implicated in AD pathology through the promotion of amyloid precursor protein processing and amyloid-β deposition[17]. Given the central role of GSK3 in these diseases, small-molecule inhibitors have been developed with some advancing into clinical trials[18]. However, these efforts have largely failed, it is believed due to either on-target based dose limiting toxicity of complete dual inhibition of GSK3α and GSK3β and/or off-target effects[19–22]. Further, genetic deletion of GSK3 paralogs has been shown to produce different phenotypes in some models when compared with inhibition[23]. Despite efforts from multiple research groups[24–26], to date a rapid, potent and proteome-wide specific GSK3 degrader of chemical probe quality[27] has not been disclosed. Such a tool could enable understanding of the potential for kinetically controlled removal of one or both isoforms and its impact on the activation status of different GSK3 substrates, as well as the potential to leverage pharmacokinetic/pharmacodynamic disconnects, characteristic of degraders[28], which may differ in effect from occupancy-based inhibition in vivo.

Here, we demonstrate a potentially generalisable bifunctional molecule synthesis and screening platform, using a concept of 'orthogonally reactive linkers', to identify a chemical probe quality and CNS in vivo active GSK3 degrader directly from a single screen.

## Results

### Developing an orthogonally reactive linker synthesis and screening platform

PROTAC potency, kinetics and absorption, distribution, metabolism and excretion (ADME) profiles are all influenced by the choice of binders, linkers and conjugation chemistry used. We therefore hypothesised that a method whereby all these components could be varied simultaneously would be advantageous to hit finding. Furthermore, with new ligase binders being disclosed on a regular basis and a desire to explore bifunctional molecules with mechanisms beyond degradation, we reasoned that an approach that did not require pre-conjugation of E3 ligase-linker intermediates would be of benefit. To address this, we proposed a concept of 'orthogonally reactive linkers', molecules that contain two distinct reactive groups, each specifically engineered to selectively react with a unique counterpart in a stepwise fashion (Fig. 1a). We envisioned a streamlined 1-well, 2-step synthesis

format such that one end of the linker reacts exclusively with binder A, while the other end selectively reacts with binder B, enabling the sequential synthesis of bifunctional molecules. We aimed to achieve this with methods that could be performed in a plate-based setting, with no protecting groups required, and with a workflow that avoided column chromatography. i.e. amenable to D2B approaches.

We were interested in focusing our library design and conjugation chemistries such that we could enable the best chances of obtaining CNS active GSK3 degraders, e.g. by limiting HBDs, amides and large increases in LogP (Fig. S1A). We also designed linkers with basic centres but avoided strongly basic centres in the binders themselves. This would allow the use of strong-cation exchange (SCX) based catch and release resins to remove starting materials and by-products that may compete with the PROTAC mechanism of action in cells[7].

Based on these criteria, we identified the simplest conditions possible for both $S_N2$ and CuAAC 'click' chemistries as a first model system (Fig. 1b). The optimised conditions for this protocol used DMF as the solvent and $Et_3N$ as the base (Fig. 2a). The conditions for the sequential CuAAC step included use of THPTA, which we found was important for basic substrates. To arrive at these conditions, we had performed a proof-of-concept screening to assess the efficiency of the $S_N2$ step under various solvent, base and additive conditions (Fig. S1B).

We designed a linker set encompassing a diverse range of properties, all featuring a secondary amine as the nucleophilic component and avoiding excessive linker length and polarity as our previous experience suggests this can cause challenges with optimisation later in a project (Fig. 2b). To maximise the potential for achieving CNS activity in an uncharted property space, we primarily incorporated CRBN-recruiting binders characterised by low molecular weight, high hydrolytic stability and/or high CRBN affinity[29]. However, given the precedence of VHL-based CNS penetrant PROTACs for LRRK2[30], we also included one VHL-recruiting binder **14** (Fig. 2c).

For GSK3 targeting binders, we designed azide containing building blocks based on the previously disclosed GSK3 inhibitors PF-367[31], **19**, and CMP-47[32] **20** (Fig. 2d). Structural comparisons showed that the triazole in PF-367 engages in a π-cation interaction with R141 and that introducing a triazole at the para position of CMP-47 could enable a similar interaction or, at the very least, avoid disrupting binding affinity (Fig. S1C–F).

With all library components synthesised (see 'Supplementary Methods' in Supplementary Information for chemistry experimental detail), we generated libraries using a liquid handling system in a 96-well plate format, with identical linker sets and conditions systematically arranged according to GSK3 binder (by plate), linker (by column), and E3-binder (by row). The synthesis began with the $S_N2$ step followed by CuAAC (Fig. 2a) and plate-based ion exchange using SCX-resin to separate unreacted species from full PROTACs. The eluted products were collected, dried, resuspended in DMSO, and analysed by LCMS for UV purity (Fig. 2e, f). We opted to screen all compounds/wells as there is no defined cut-offs for what constitutes sufficient purity for screening and in fact would likely rely not only on the purity of the desired compound, but also the nature of the impurities. Nevertheless, as a guide, our analysis showed that a total of 147 compounds were detected by mass spectrometry, with 103 exhibiting >50% UV-purity across the two plates. Other groups have reported this to be a purity level at which minimal impact on assay interpretation is expected[10]. In addition, we observed that non-benzyl bromides are potentially less effective than benzyl bromides in the $S_N2$ reaction, and that spirocyclic amine-based linkers displayed poor success rates, informing on potential areas for future optimisation of the approach. Taken together these data, representing the simultaneous variation of multiple GSK3 binders, E3 ligase binders and linker types demonstrate the effectiveness and utility of the proposed orthogonally reactive linker concept for bifunctional molecule library construction.

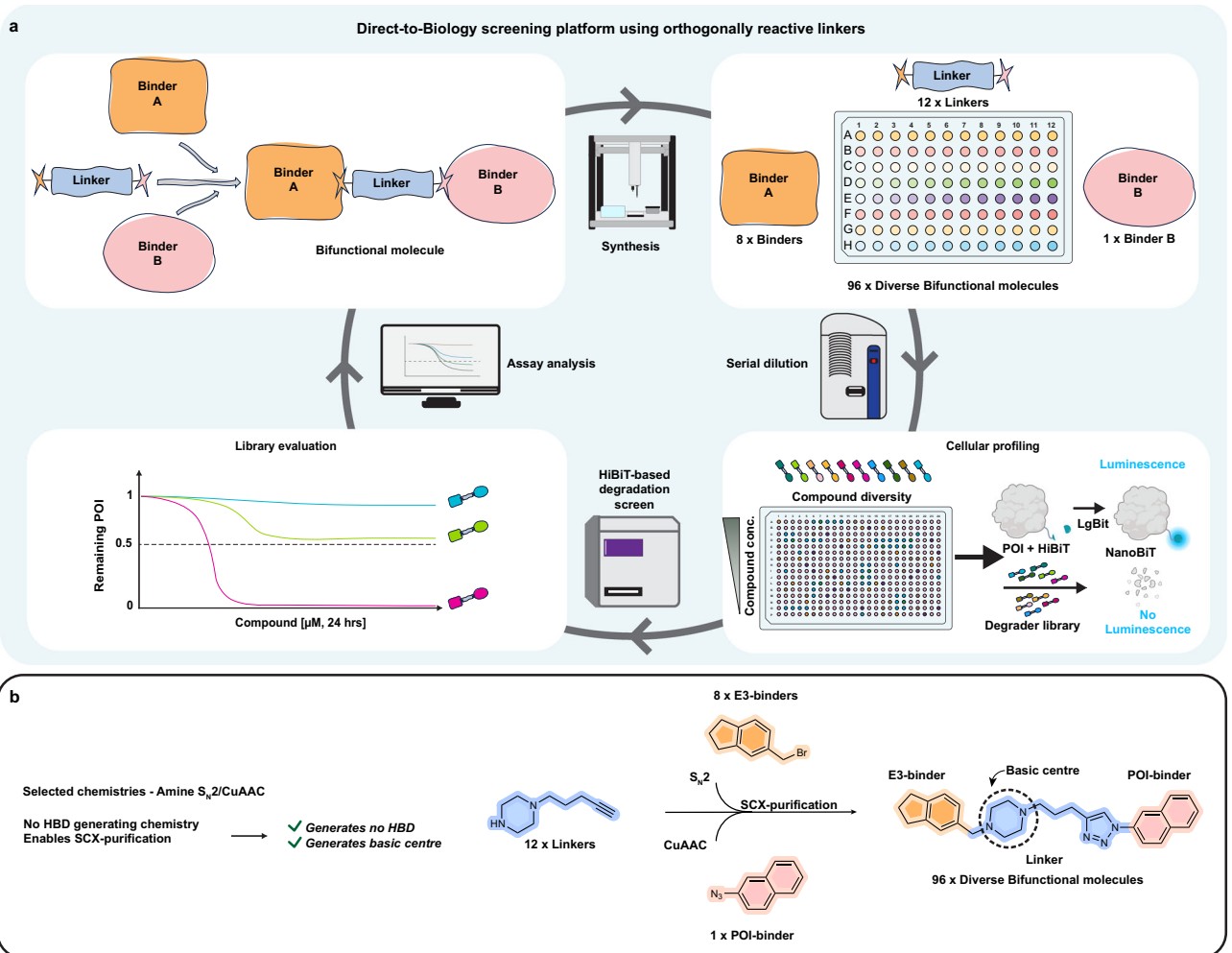

**Fig. 1 | An orthogonally reactive linker synthesis and screening platform concept. a** Proposed approach for simultaneous variation of all components of a bifunctional molecule in a library setting with direct-to-biology cellular screening against a protein-of-interest (POI). Created in BioRender. Farnaby, W. (2025) https://BioRender.com/ncfcg8e. **b** Selected conjugation chemistries, followed by strong cation exchange (SCX) yields direct-to-biology (D2B) libraries. HBD hydrogen bond donors.

## Identifying GSK3 degraders from a direct-to-biology screen

To enable library screening, we generated endogenously tagged GSK3β-HiBiT knock-in (KI) HEK293 cells (Fig. S2A–C). Treatment of GSK3β-HiBiT KI HEK293 cells with cycloheximide showed that rates of GSK3 decay were long (>24 h) and not affected as compared with wild-type (WT) HEK293 cells over 48 h (Fig. S2D, E). Crude compound mixtures, initially made to 10 mM DMSO stock solutions on the assumption of 100% conversion and purity, were taken directly into cell-based high-throughput degradation screens with 24 h treatment. Dose-dependent degradation profiles and $DC_{50}$ heatmaps of library plate 1 (Fig. 3a, c) and library plate 2 (Fig. 3b, d) show that compounds from both libraries degrade GSK3β to various extents. Importantly, we included in both plates a negative control row, utilising benzyl bromide as a mock-E3 ligase reagent (Fig. 2c, e, f, row e in both plates). Pleasingly, all wells from negative control compounds showed no effect on GSK3β-HiBiT signal, confirming that the chemistry we used did not interfere with the assay read-out (Fig. 3a–d). PT-65 is a previously reported GSK3 degrader and was used as a benchmark in our assays[25] (Fig. 3b). From this single screen, structure-activity based relationships were generated, revealing that regardless of the GSK3-binder choice, some linkers (e.g. **2** and **4**) as well as E3 ligase binders (**16** and **17**) showed a trend towards more effective GSK3β degradation. GSK3-binder **20** used in the generation of library plate 2 resulted in more potent compounds. In fact, 6

compounds from plate 1 demonstrated >50% $D_{max}$ and <1 μM $DC_{50}$, whereas 35 compounds from plate 2 met the same criteria. Next, we sought to further characterise 6 of the selected compounds from library plate 2. Considering both potency and a desire to further profile compounds with a mix of different linkers and E3 ligase binders we chose compounds **21**, **22**, **23**, **24**, **25** and **26**, which originated from wells C7, F2, F12, D2, D7 and D12, respectively. The selected compounds were re-synthesised and purified via preparative HPLC and the degradation profiles of crude vs purified compounds assessed (Fig. 3e–j). Observed $DC_{50}$ values were left-shifted (2.5–20 fold shift) for all selected pure compounds, whilst $D_{max}$ values remained the same between crude and pure compounds. Approximate rank ordering also remained the same amongst the set and we expect the degree of $DC_{50}$ shift between crude and pure compounds reflects the actual vs assumed (10 mM) concentration of each compound in the crude stock solution produced for testing, as well as the impact of any remaining competing impurities. We were able to confirm that all compounds, once resynthesised, retained their high levels of degradation potency. All compounds displayed $DC_{50}$ values ≤20 nM and compound **26** yielded sub-nanomolar potency. PROTAC-mediated degradation of untagged, endogenous GSK3β in HEK293 cells with 24 h treatment was also confirmed via immunoblotting (Fig. S2F, G), assuring us that all compounds are bone fide GSK3 degraders.

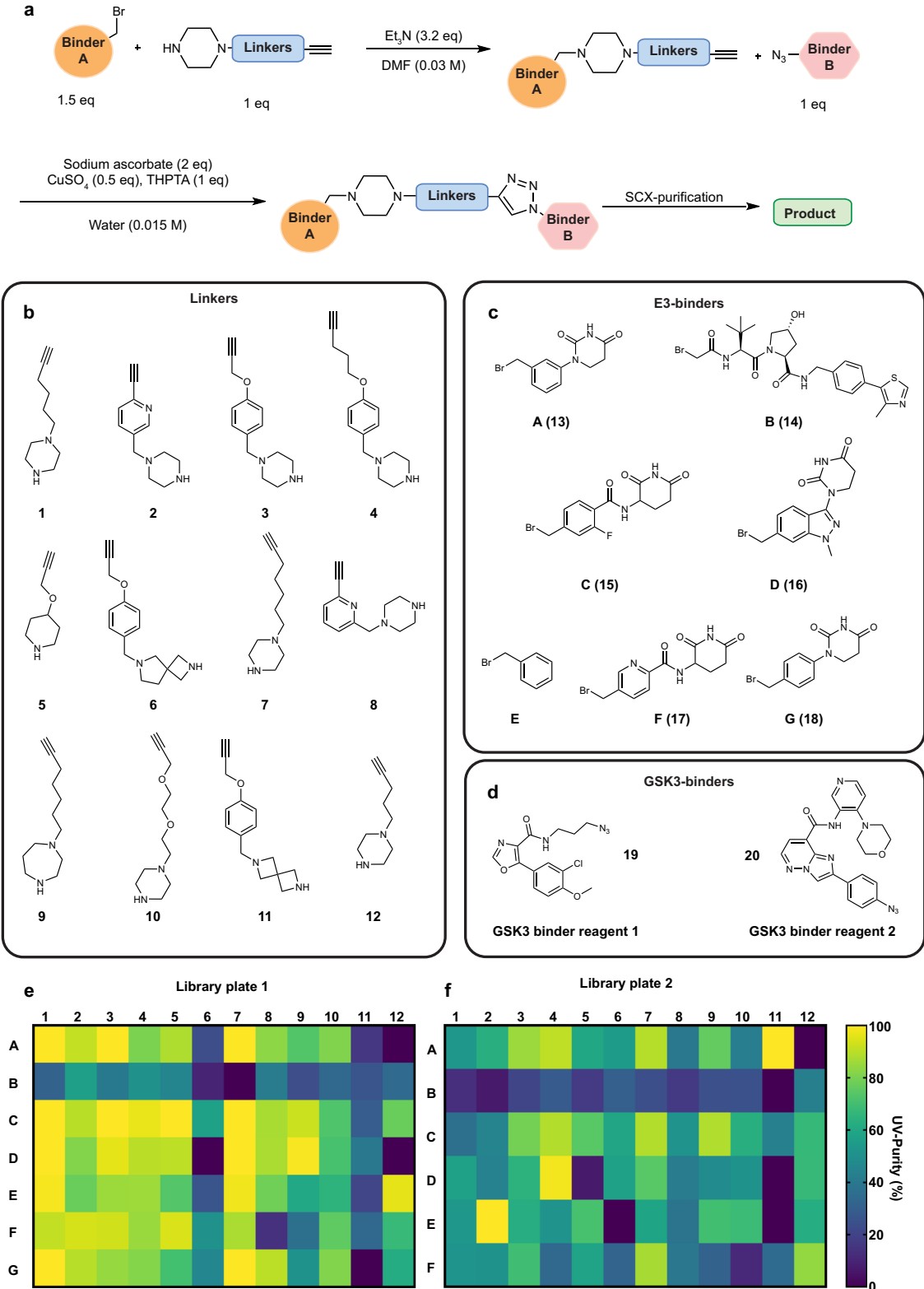

**Fig. 2 | Synthesis of a PROTAC screening library using orthogonally reactive linkers and S$_N$2/CuAAC chemistry. a** Selected reaction conditions for the D2B S$_N$2/CuAAC chemistry **b** design of orthogonally reactive linker reagents to cover a broad physicochemical property space, including aromatic, heteroaromatic, rigid, PEG-based and aliphatic linkers. **c** E3-ligase binder building blocks. **d** Selected GSK3-targeting binders containing azide functionalities for CuAAC conjugation. **e** Library Plate 1: PROTAC library based on binder **19**, systematically arranged by linkers in columns and E3-binders in rows, with heat-maps based on UV-purity data. **f** Library Plate 2: PROTAC library using binder **20**, with a similar reaction layout and organisation as in Library Plate 1.

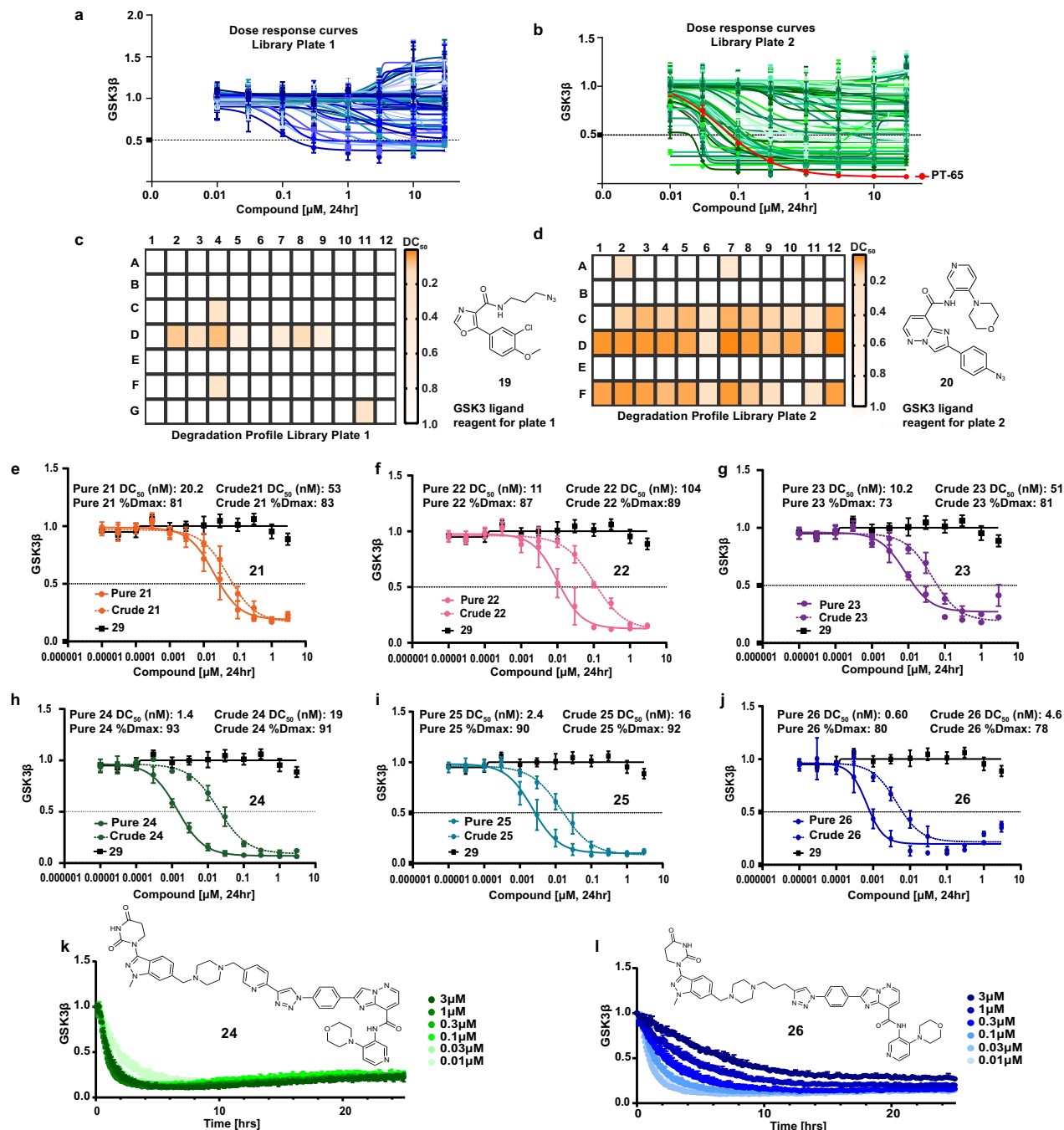

**Fig. 3 | Identifying GSK3 degraders from a direct-to-biology screen. a–d** HiBiT lytic assay-based high-throughput screening for GSK3β abundance with 24 h PROTAC treatment in GSK3β-HiBiT HEK293 cells. Dose response curves were generated with non-linear 4 parameter fit correction, and representative graphs are combined results of $n = 2$ biological replicates, each from $n = 2$ technical replicates with error bars representing mean and ± SD (**a** shades of blue correspond to individual compounds from Library Plate 1; **b** Shades of green correspond to individual compounds from Library Plate 2, with the red curve indicating the positive control PT-65). Heatmaps were generated using $DC_{50}$ values. Compounds with $D_{max} < 30\%$ were considered as having $DC_{50} > 1 \mu M$. **c, d** Shading intensity reflects compounds $DC_{50}$ values, with white indicating >1 μM and deep orange indicating <100 nM. **e–j** Compounds **21** (orange), **22** (pink), **23** (purple), **24** (green), **25** (teal) and **26** (blue) were selected for resynthesis and further characterisation, which were

originated from wells C7, F2, F12, D2, D7 and D12 of library plate 2, with the GSK3 inhibitor **29** (black) included as a negative control for comparison. Potency of these purified compounds were compared with crude mixtures (dashed lines) for GSK3β degradation upon 24 h treatment in GSK3β-HiBiT HEK293 cells (Dose response curves are means of $n = 3$ biological replicates ± SD, 4-parameter non-linear curve fitting (GraphPad). **k, l** Kinetic live cell degradation of GSK3β with selected compounds **24** (green) and **26** (blue), respectively, monitored in LgBit overexpressing GSK3β-HiBiT KI HEK293 cells where shading intensity reflects compound concentration, with darkest shading indicating highest tested concentration (3 μM) and lightest shading indicating lowest tested concentration (0.01 μM) ($n = 3$ biological replicates, and data representatives of one biological replicate per compound with error bars representing mean of technical replicates and ± SD).

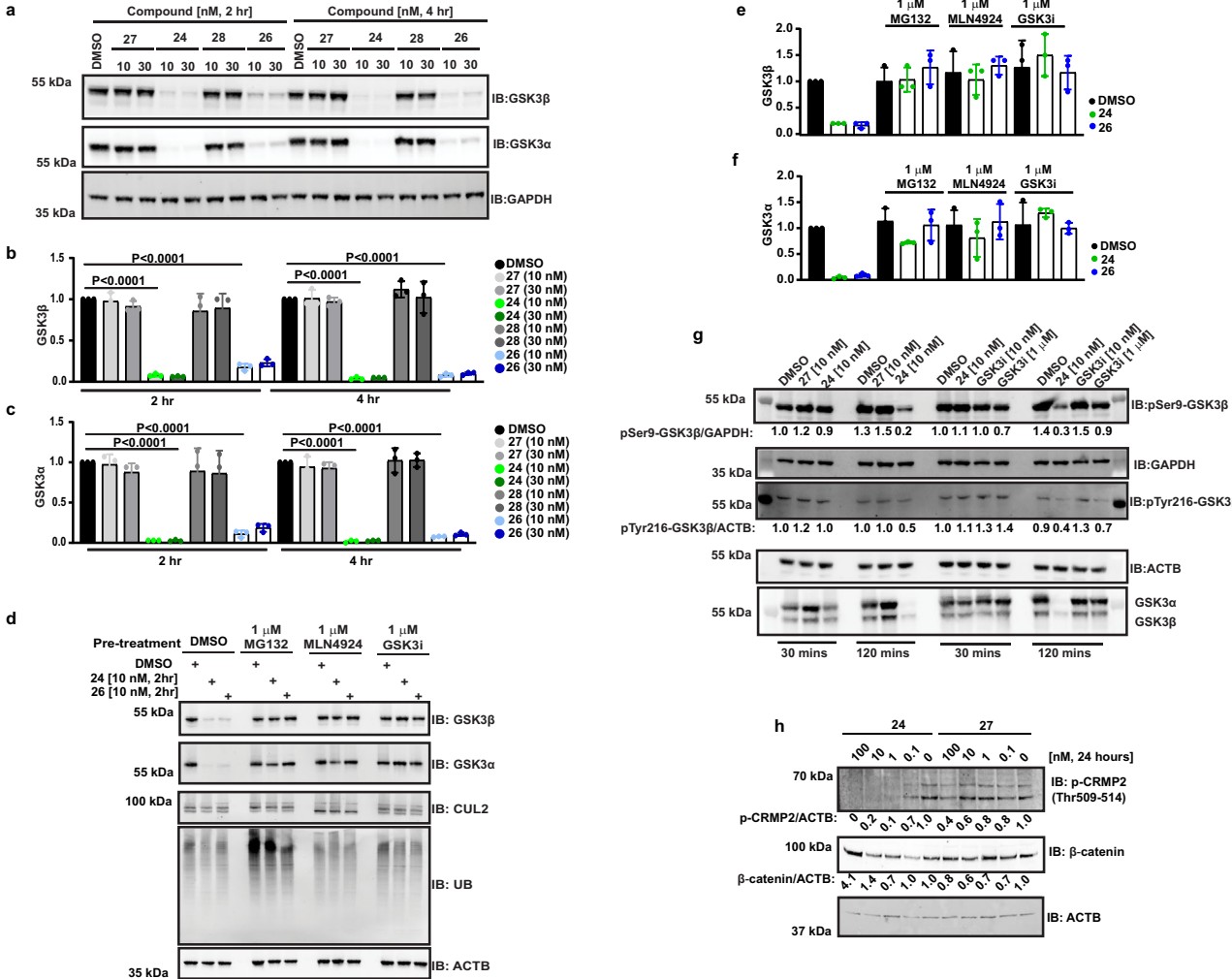

**Fig. 4 | Characterisation of cellular GSK3 degrader probes. a** Endogenous GSK3 paralog degradation with compounds **24** and **26** and negative controls **27** and **28** in HEK293 cells (representative blots of *n* = 3 biological replicates, 2–4 h). The samples derive from the same experiment, gels and blots were processed in parallel. **b, c** Immunoblots shown in panel A were quantified, normalised to relevant internal controls and plotted for changes in GSK3β and GSK3α levels. Compound **27** in light grey, **24** in green, **28** in dark grey, and **26** in blue, with shading intensity corresponding to concentration (lightest = 10 nM, darkest = 30 nM) (*n* = 3 biological replicates with error bars representing mean and ± SD, *p* values are calculated as <0.0001, two-tailed unpaired *t*-test (GraphPad)). **d** Mechanism of action studies in HEK293 cells for compounds **24** and **26** showing rescue with 2 h pre-treatment of proteasome inhibitor (1 μM, MG132), neddylation inhibitor (1 μM, MLN4924) and

GSK3 inhibitor (1 μM, **29**). Sample processing controls run on different gels in parallel (Representative blot of *n* = 3 biological replicates). **e, f** Immunoblots shown in (**d**) were quantified, normalised to relevant internal controls and plotted for changes in GSK3β and GSK3α levels, **24** in green and **26** in blue (*n* = 3 biological replicates with error bars representing mean and ± SD). **g** Evaluation of changes in phosphorylation levels of GSK3β following compound **24**, **27** or GSK3 inhibitor (**29**) treatments for 30 and 120 min in HEK293 cells (*n* = 3 biological replicates). The samples derive from the same experiment, gels and blots were processed in parallel. **h** Dose dependent effect of compound **24** on GSK3 substrates in differentiated SH-SY5Y cells (representative of *n* = 3 biological replicates, at indicated concentrations, 24 h). The samples derive from the same experiment, gels and blots were processed in parallel.

Next, we wanted to investigate the degradation rates of the six selected compounds. We used LgBit transfected GSK3β-HiBiT cells to monitor live cell dose-dependent degradation over time. Maximal degradation rates ($\lambda_{max}$) varied between test compounds (Figs. 3k, l and S2H–L), with an observed maximal degradation rate for **24** of -1.0 h⁻¹ (Fig. S2L). Due to the hook effect, maximal degradation rates couldn't be calculated for some of the compounds over the concentration range used, including **26**. However, almost complete degradation of GSK3β was observed for **26** after 2 h with as little as 10 nM compound concentration. We were also able to demonstrate no impact on cell viability in this cell line up to 24 h and 1 μM treatment of GSK3 degrader using compound **24** (Fig. S3A–F).

All together, these data highlighted that from one orthogonally reactive linker D2B screen, we could identify potent and rapid GSK3 degraders from multiple series.

**Evaluation of cellular GSK3 degrader probes**

Based on the kinetic degradation profiling, we focussed further on compounds **24** and **26**. We evaluated endogenous degradation of both GSK3α and GSK3β at short time points in HAP1 and HEK293 cells and compared performance to negative control compounds (**27** and **28**) that are unable to engage CRBN, due to methylation of the Dihydrouracil ring (Figs. 4a–c and S4A–E). In line with the live cell degradation data, almost complete GSK3α and GSK3β degradation was achieved with 10 nM treatment for 2 h with **24** and **26** (Fig. 4a–c). Overall, we observed no appreciable specificity for individual paralogs with these two PROTACs.

Follow-up mode-of-action studies showed that degradation was rescued in the presence of proteasome inhibitor (MG132), nedd8-activating enzyme inhibitor (MLN4924) and GSK3 inhibitor/competitor of GSK3, compound **29** (Fig. 4d–f), supporting that **24** and **26**

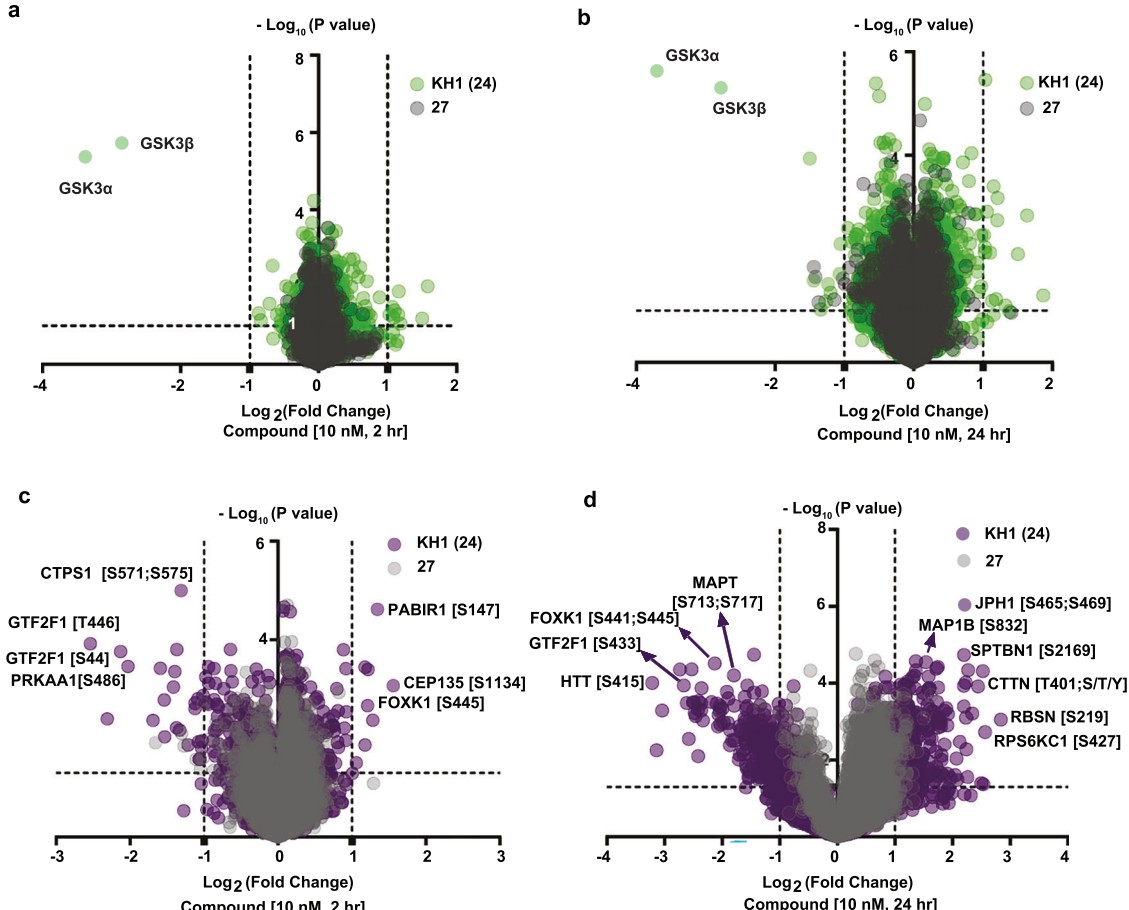

**Fig. 5 | Global and phospho-proteomic analysis of KH1 (24).** Proteome wide compound selectivity for compound **24** (green) vs **27** (grey) normalised to DMSO, following 10 nM treatment for 2 h in (**a**) and 24 h in (**b**) in HEK293 cells ($n = 3$ biological replicates). Phosphoproteomic analysis of compound **24** (purple) and **27** (grey) normalised to DMSO following 10 nM treatment for 2 h in (**c**) and 24 h in (**d**) in HEK293 cells ($n = 3$ biological replicates). Two-tailed $t$-test performed with 250 randomisations to assess the statistical significance between groups using Perseus software.

mediated GSK3 degradation requires UPS activity and GSK3 engagement. Next, we compared GSK3 degradation in HAP1 WT and CRBN knock-out cells. We observed rescue in degradation confirming CRBN dependency of PROTAC-mediated degradation (Fig. S4F–H). To ascertain order of events, we also evaluated the impact on GSK3 activity prior to degradation (Fig. 4g). Here we show that following treatment of HEK293 cells with 10 nM **24**, negative control **27** and both 10 nM and 1 μM GSK3 inhibitor (**29**), there is no impact observed on either GSK3β Ser9 (an inhibitory site) or GSK3β Tyr216 (an autophosphorylation activating site). Changes are only appreciably observed in total and phosphorylated GSK3β levels concurrently once **24** mediated degradation ensues at 2 h treatment, as well as some reduction in phosphorylation of Tyr216 after 2 h for the 1 μM GSK3 inhibitor (**29**) treatment. This is to be expected from a catalytic, non-occupancy-based PROTAC mechanism of action.

We also investigated the effect of **24** on GSK3 and GSK3 substrates in differentiated SH-SY5Y neuroblastoma cells (Figs. 4h and S4I). We focussed on assessing impact of GSK3 degradation on total β-catenin and phospho-Collapsin response mediator protein 2 (CRMP2) levels. We chose β-catenin due to the established role of GSK3 in promoting its degradation via phosphorylation[33] and the proposed on-target toxicity risks for GSK3 inhibitors that have been observed to stabilise β-catenin. In addition, CRMP2 is a validated neuronal GSK3 target, which has been found to be hyperphosphorylated in the brains of AD patients and suggested to be an early event in AD pathogenesis[34]. With 24 h compound treatment the onset of GSK3 degradation was observed at between 0.1 and 10 nM with concurrent reduction in phospho-CRMP2,

with impact on β-catenin stabilisation most clear at the highest tested dose of 100 nM.

Next, to gain further insight into the proteome-wide specificity of our compounds, HEK293 cells were treated with 10 nM of active compounds **24** and **26**, and related negative controls **27** and **28** and DMSO (vehicle control) for 2 h. Tandem-Mass-Tagging based global proteomics data showed exquisite selectivity for GSK3 paralogs for **24** when compared to DMSO and to negative control **27** (Fig. 5a and Supplementary Data 1). A similar selectivity profile was observed for compound **26** in comparison with DMSO and to negative control **28** (Fig. S4J and Supplementary Data 2). To understand what changes to the whole proteome may occur over time following GSK3 degradation, we also performed the same experiments with extended compound treatment times of 4 and 24 h (again at 10 nM) (Figs. S4K and 5b, Supplementary Data 3 and 4). These experiments showed consistent proteome-wide selectivity for GSK3α and GSK3β in HEK293 cells and little alteration to abundance of proteins across the proteome following treatment with **24**, when compared to DMSO and negative control **27** even at 24 h (Fig. 5b). We did observe a greater number of proteins with increased abundance at the 24 h time point, including a small increase in β-catenin at both the 4 and 24 h time-points (4 h = $\log_2$ fold change of +0.63, $p < 0.0005$; 24 h = $\log_2$ fold change of +0.89, $p < 0.005$) broadly in line with observations at the same dose and time-point in SH-SY5Y cells. Having observed the rapid and specific nature of **24** mediated GSK3 removal, we evaluated whether this tool molecule could be used to shed light on the GSK3 regulated phosphoproteome. Given our data

supporting little impact on GSK3 levels following treatment of HEK cells with 10 nM **24** at 30 min, but almost complete removal by 2 h (Fig. 4g), we opted to enrich phosphopeptides from the same 2, 4 and 24 h samples used for whole proteomic analysis and conduct unbiased mass spectrometry quantitation to understand the change in the phosphoproteome over time (Figs. 5c, d, S4L and Supplementary Data 5–7). Previously, a number of bone fide GSK3 substrates have been validated[14], with many more potential substrates identified following genetic removal of GSK3 paralogs[35]. Here, however, **24** mediated removal and subsequent phosphoproteomic analysis gave a first glimpse of the kinetic nature of substrate vulnerability in a manner not previously achievable. For example, at early time points of 2 and 4 h, whilst little change was observed, some previously reported substrates of GSK3 were shown to be rapidly dephosphorylated in the absence of GSK3, such as AMPK alpha subunits (e.g. PRKAA1)[36], whereas other well profiled and neurodegenerative disease relevant substrates such as Tau (MAPT)[37] and Huntingtin (HTT)[38] were dramatically impacted, but not until the later 24 h time-point (~22 h post-GSK3 complete removal).

Here we show that both compounds **24** and **26**, hereafter referred to as KH1 and KH2, respectively, are fast, low to sub-nanomolar degraders of both GSK3 paralogs and that KH1 demonstrated significant impact on CNS relevant GSK3 substrates in cells. They demonstrate very high levels of proteome-wide specificity, qualifying them as cellular chemical probe quality GSK3 degrader molecules identified from a single D2B screen. In addition, we were able to demonstrate the utility of KH1, for investigating downstream consequences of GSK3 protein removal over time, which we envisage will be of particular use in the future for interrogating more complex and disease-relevant models.

## KH1 is a brain active GSK3 degrader

Given the high levels of on-target potency and specificity, we were interested to evaluate the performance of some of our compounds with respect to their in vitro and in vivo pharmacokinetic profiles. **21**, **22**, KH1 (**24**), **25** and KH2 (**26**) were all tested for liver microsome stability and kinetic solubility (RealSol, pH 7.4) (Supplementary Table 1). As is often the case for bifunctional molecules, solubility was low but detectable in the low micromolar range for compounds **21**, KH1 (**24**) and KH2 (**26**). Liver microsome data were variable across the compound set and **22**, KH1 (**24**) and KH2 (**26**) showed low-to-moderate scaled microsomal clearance across species (Supplementary Table 1). We therefore moved forward to investigate in vivo clearance in a mouse cassette PK experiment using the better performing compounds KH2 (**26**) and **22** as well as including **25** to check the in vitro/in vivo clearance relationship (Fig. S5A and Supplementary Table 2). Following i.v. dosing (0.5 mg/kg) intrinsic plasma clearance of **22** and **26** was shown to be moderate with elevated clearance for **25**, consistent with the in vitro data. KH1 (**24**) was previously shown to be one of our most potent hits and shows a diminished hook-effect in cellular degradation assays as compared with other analogues such as KH2 (**26**) (Fig. 3j, l). Furthermore, in vitro microsomal stability and solubility data for KH1 (**24**) were comparable with KH2 (**26**), which performed well in the in vivo PK cassette study. We therefore selected compound KH1 (**24**) for a discrete i.v./p.o. PK study, measuring both plasma and brain concentrations (Fig. 6a and Supplementary Tables 3 and 4). KH1 (**24**) displayed low, but measurable, oral bioavailability of 1.6% and a total mean plasma concentration above 65 nM for up to 4 h following i.v. dosing at 0.37 mg/kg. Whilst we measured exposure up to 24 h, due to the rapid degradation kinetics of KH1 (**24**) we were particularly interested to understand compound exposures at the shorter time points. Notably, we also observed some penetration of compound KH1 (**24**) into the brain, with a total concentration of 16 nM at 2 h and a brain: plasma ratio of 0.18 at this time point (Supplementary Table 4).

Given the excellent potency, rapid degradation kinetics and observed in vivo exposures of KH1 (**24**), we were interested to assess the potential for in vivo activity. We first looked to predict the free compound concentrations. Assessing the extent of plasma protein binding has proven challenging for bifunctional molecules due to often being very highly plasma protein bound and at the upper limits of accurate quantitation using classical assay techniques[28]. We therefore took a dual approach, measuring plasma protein binding via a standard equilibrium dialysis assay (Supplementary Table 1) and in addition by measuring degradation of GSK3β-HiBiT in the presence of 10% foetal bovine serum or 10% mouse serum for KH1 (**24**) (Fig. 6b) and KH2 (**26**) (Fig. S5B), as has been described previously[28]. Equilibrium dialysis suggests an unbound fraction of 0.5% for KH1 (**24**) in mouse plasma. In reasonable agreement with this, the serum shift degradation assay showed a shift in $DC_{50}$ of 8.2× with 10% Mouse serum (Fig. 6b), therefore predicting a shift of ~80× in whole blood and equating to a free fraction of ~1.2% in mouse plasma, giving us extra confidence for predicting free exposure in vivo to guide our dosing regimen. Using a 2-h total plasma concentration of 88 nM at a dose of 0.37 mg/kg (Supplementary Table 4) and using the predicted free fractions, an unbound total plasma concentration at this time point of between 0.4 and 0.9 nM was calculated. In addition, we measured brain binding, predicting a free concentration in brain of 20 pM based on the measured free fraction of 0.12%. This equates to a low estimated Kp,uu of 0.05. Finally, to understand if KH1 (**24**) could reasonably be expected to degrade GSK3 in vivo and at what dose, we measured the ability of KH1 (**24**) to degrade mouse GSK3β in mouse embryonic fibroblasts (Fig. 6c, d), suggesting a $DC_{50}$ in the region of 0.1–1 nM. We therefore reasoned that although the Kp,uu was low, the high potency of KH1 (**24**), its measured brain concentration/penetration and total plasma levels could reasonably mean it may reach required free concentrations to observe effects on GSK3 abundance in vivo at a ~10-fold elevated i.v. dose (i.e. in the region of 3–5 mg/kg).

Given this understanding we dosed mice at 5 mg/kg i.v. and measured KH1 (**24**) levels in plasma and brain at 4 h, as well as GSK3β abundance in liver and brain at the same time point. Pleasingly, we observed almost complete removal of GSK3β in liver as well as significant degradation of GSK3β in the mouse brain (Fig. 6e, f). We also conducted mass spectrometry-based analysis of changes to the whole proteome and phosphoproteome on the same liver tissue samples, as well as the impact on beta catenin and p-CRMP2 levels via western blot (Figs. 6g, h, S5C, D, and Supplementary Data 8 and 9). Consistent with our cellular data (Fig. 5) we observed high levels of proteome wide specificity for GSK3β removal (GSK3α could not be quantified) in the liver and limited but notable impact to the phosphoproteome at this time-point, likely a time-point at which GSK3 has not long been removed. For example, we observed changes to phosphorylation of the AMPK alpha sub-unit (PRKAA1), which was again consistent with our early time-point HEK293 data, as well as a reduction in GSK3α Ser21 phosphorylation in line with the expected reduction in total GSK3α.

## Discussion

In this study, we present a library synthesis and screening approach that was able to identify an in vivo active bifunctional degrader of chemical probe quality directly from the initial screen. We started by establishing use of linker reagents that can be selectively conjugated in high conversion, with two different reaction types at either end, so-called orthogonally reactive linkers. Specifically, we used a combination of $S_N2$ and CuAAC reactions, choosing these reaction types due to their potential for high conversion, whilst limiting the extra polar surface area and HBD contribution. A key design feature was the inclusion of a basic centre in the linkers, allowing us to perform a plate-based SCX step prior to compound testing to remove any unreacted binder reagents and related by-products, which had been observed as a challenge in previous studies[7]. This led to almost all attempted

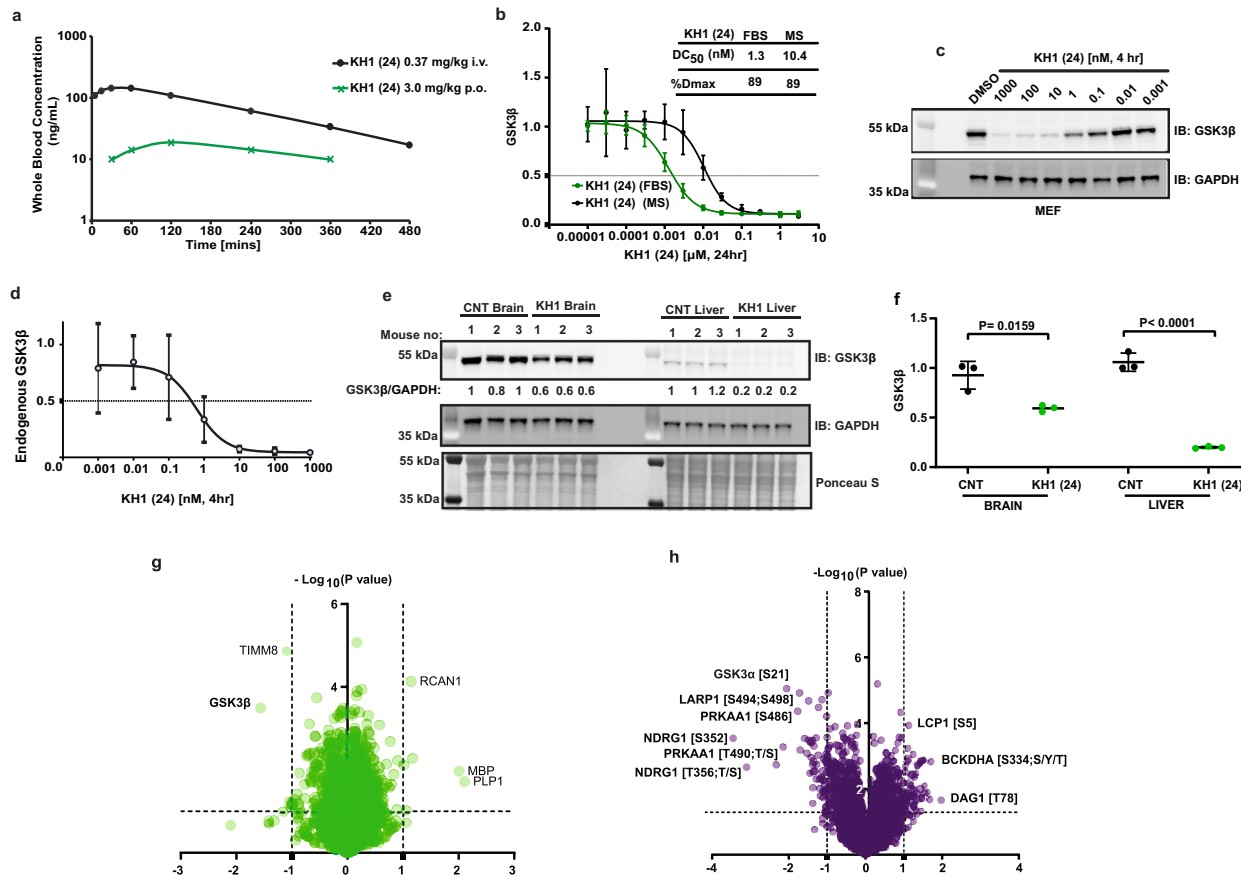

**Fig. 6 | In vivo PK/PD profiling of KH1 (24). a** Pharmacokinetic profile of KH1 (**24**) in female Balb/c mice following a single intravenous (i.v.) bolus dose of 0.37 mg/kg (black dot) or oral (p.o.) gavage dose of 3.0 mg/kg (green cross). Mean of samples from three mice per dose route showing whole blood concentration-time profile. **b** Serum shift assay. Degradation of GSK3β-HiBiT by KH1 (**24**) in GSK3β-HiBiT KI HEK293 cells in presence of 10% foetal bovine serum (FBS) (green) or mouse serum (MS) (black) (24 h, *n* = 3 biological replicates, with error bars representing mean and ± SD, 4-parameter non-linear curve fitting (GraphPad)). **c** Immunoblotting based endogenous degradation profile of KH1 (**24**) in mouse embryonic fibroblast (MEF) cells for 4 h with indicated concentrations (Data representative of *n* = 3 biological replicates). **d** Dose dependent GSK3β degradation of KH1 (**24**) in MEF cells (4 h, *n* = 3 biological replicates, with error bars representing mean, ± SD, 4-parameter non-linear curve fitting (GraphPad)). **e** Assessment of GSK3β in liver and brain following single i.v. dose of KH1 (**24**) 5 mg/kg in female Balb/c mice after 4 h, data shows samples from three mice compared to control (CNT). **f** Relative quantification of GSK3β levels in brain and liver from (**e**) following a single i.v. bolus dose of 5 mg/kg of KH1 (**24**) (green) compared to control (CNT) (black) (*n* = 3 mice, error bars representing mean and ± SD, two-tailed unpaired *t*-test, *p* values calculated as *p* = 0.0159 for brain samples and *p* < 0.0001 for liver samples (GraphPad)). **g** Whole Proteome (green) and **h** Phospho-proteome (purple) analysis of liver tissue samples (*n* = 3 mice) collected from the pharmacodynamic study as detailed in (**e**). Two-tailed *t*-test performed with 250 randomisations to assess the statistical significance between groups using Perseus software.

PROTACs being detected via LCMS and the vast majority of failures came from either the use of an alpha-bromoacetamide electrophile or linkers containing spirocyclic motifs. This highlights both the generally robust nature of the protocol and opportunities for further optimisation.

Next, we tested our library in a GSK3β-HiBiT knock-in HEK293 cell line, generating full dose-response curves on all compounds. Once all building blocks and a suitable screening assay were available, we were able to complete the full synthesis and screening workflow in 5 days. This rapidly led us from a position of no-prior in-house GSK3-related degrader data to identifying a large number of hits, with focus falling on six molecules that were resynthesised and purified for further profiling. Of particular note, we identified two molecules, KH1 (**24**) and KH2 (**26**), that showed DC$_{50}$ values in the picomolar to single-digit nanomolar range and exquisite specificity for degrading both GSK3 paralogs with no other significantly degraded proteins identified after 2 h treatment. To support the utility of these probes in understanding GSK3-related pathway biology, we investigated their impact on established GSK3 substrates β-catenin and phospho-CRMP2[33,34] in differentiated neuroblastoma SH-SY5Y cells, as well as performing both whole proteome and phosphoproteomic analysis across a series of timepoints in HEK293 cells. Initial data suggest the AD related phospho-CRMP2 is reduced in line with GSK3 degradation. We also observed that some established GSK3 substrates, such as AMPK, were impacted rapidly after and/or along with GSK3 removal, with phosphorylation of other disease-relevant substrates, such as Tau, appearing to be impacted but only at the later 24 h time-point, giving a first look at the kinetics of substrate vulnerability to GSK3 removal. This data highlights the opportunities that are now enabled to further interrogate how rapid removal of GSK3 paralogs may impact the activity, particularly kinetically, of disease-relevant substrates as compared with those that may be deemed to be safety risks, and how this compares to inhibitors in more complex disease models. Finally, we discovered that KH1 (**24**), following a single 5 mg/kg i.v. dose, was able within 4 h to almost completely remove GSK3β from mouse liver with partial degradation also observed in mouse brain. Our data support that this is achieved via a combination of moderate brain penetration and potent and rapid degradation kinetics. This study shows that the inherent pharmacokinetic/pharmacodynamic disconnect of PROTACs can be leveraged to avoid the requirement for high and

prolonged CNS exposure, which would be required for inhibitors, to achieve on-target modulation in the brain. Whilst an opportunity for some, this may also act as an important warning to those who require avoiding on-target activity in the brain. We hope that the orthogonally reactive linker synthesis and screening approach exemplified here can provide a generalisable strategy to accelerate bifunctional probe discovery and therapeutic concept generation and a benchmark for CNS PROTAC discovery.

## Methods

### Chemical synthesis
Synthesis experimental detail is provided in 'Supplementary Methods' as part of the Supplementary Information file.

### Cell culture
HEK293 cells purchased from ATCC and MEF cells were kindly gifted by MRC PPU reagents and services at the University of Dundee. HEK293 and MEF cells were maintained in DMEM (high glucose, GlutaMAX; Gibco) containing 10% foetal bovine serum (FBS; Thermo Fisher), 100U ML-1 penicillin-streptomycin (Thermo Fisher). HAP1 wild type and HAP1 Cereblon Knock-out cell line (CRBN-KO, HZGHC003752c001, Horizon) was a gift from the Ciulli Lab. HAP1 cells were cultured in IMDM (Gibco) with the same additives described as above. SH-SY5Y human neuroblastoma cells (ATCC, CRL-2266) maintained in DMEM (high glucose, Gibco) supplemented with same additives as above. For differentiation, $5 \times 10^5$ cells were seeded into each well of 6 well plates and differentiated into a more neuronal-like phenotype by treating for 5 days with $10 \mu M$ retinoic acid (F9665, Sigma), changing the media every day for the first 3 days[39,40]. All cell lines were authenticated (Eurofins UK, www.eurofins.co.uk) and STR validation reports are provided for download (Supplementary Data 10). All cells were cultured in a humidified incubator at $37 \degree C$ and 5% $CO_2$ and routinely tested for mycoplasma contamination. All cell culture procedures were performed under aseptic conditions in line with biological safety requirements.

### CRISPR/Cas9 mediated HiBiT Knock-in cell line generation
GSK3β-HiBiT HEK293 cells were generated via RNP transfection method of delivering single-stranded oligonucleotides (IDT) as the Alt-HDR donor template (IDT, Ref#236644298), spCas9 (Sigma-Aldrich) and target-specific Alt-R CRISPR-Cas9 crRNAs (IDT, Ref#236644305, 236644306 and 236644307). Briefly, two parts of the guides (IDT, trRNA and crRNA) were mixed according to manufacturer's instructions and annealed at $95 \degree C$ for 5 min to form gRNA, after gRNAs were cooled down, Cas9 was added to the mixture to form an RNP complex. HEK293 cells were resuspended in R buffer and RNP complex added in to the suspension alongside with donor template and cells were electroporated using a Neon Electroporation system (Thermo Fisher). After electroporation, cells were plated in six-well plates in pre-warmed culture media. After 48–72 h of post electroporation, knock-in efficiency of the pool cells was tested by via HiBiT lytic detection system (Promega) using PheraStar plate reader (BMG). After successful detection of HiBiT signal, the HiBiT KI pool population underwent single-cell sorting (Sony Biotechnology, SH800). After 2–3 weeks of recovery in conditioned media and expansion, validation of HiBiT insertion of single cell clones was conducted via HiBiT lytic assay (Promega), western blotting and finally amplicon sequencing (MiSeq, Illumina).

### Amplicon sequencing
Genomic DNAs from cell pellets were extracted using QuickExtract DNA extraction solution (Lucigen Catalog #QE09050) according to the manufacturer's instructions. Amplicon libraries were generated using a two-step PCR protocol. In the first PCR, target-specific primers with Illumina overhang adaptor sequences (Forward 1: TCGTCGGCAGCGTCAGATGTGTATAAGAGACAGC AAACGTGACCAG TGTTGCT, Reverse 1: GTCTCGTGGGCTCGGAGATGTGTATAAGAGA-CAGT GCTCATGCTCTCTCTTGGG, Forward 2: TCGTCGGCAGCGT-CAGATGTGTATAAGAGACAGCTT TCCAAACGTGACCAGTGT and Reverse 2: GTCTCGTGGGCTCGGAGATGTGTATAAGAGA CAGG CTC ATGCTCTCTCTTGGGAC) were used to amplify the region of interest. PCR reactions were performed in $50 \mu L$ volumes using Platinum™ SuperFi II PCR Master Mix (Thermo Fisher Scientific Catalog no: #12368010) with 2× buffer, $0.5 \mu M$ each primer, and $5 \mu L$ of template DNA (genomic DNAs from QuickExtraction). Thermal cycling conditions were as follows: initial denaturation at $98 \degree C$ for 30 s; followed by 30 cycles of $98 \degree C$ for 10 s, $60 \degree C$ for 20 s, $72 \degree C$ for 30 s; and a final extension at $72 \degree C$ for 5 min. Amplicons were purified using AMPure XP beads (Beckman Coulter) at a 1:1 ratio and eluted in $30 \mu L$ of nuclease-free water. For the indexing PCR, dual-index barcodes were added using Illumina indexing primers (Illumina DNA/RNA UD Indexes Set A, Tagmentation (20091654). The Indexing PCR was performed with 8 cycles of amplification using KAPA HiFi HotStart ReadyMix (Roche), followed by another round of AMPure XP bead purification (1:1) and elution in $20 \mu L$ 10 mM TRIS pH8.5. Purified libraries were quantified using the Qubit dsDNA HS Assay Kit (Thermo Fisher Scientific) and library size was verified by capillary electrophoresis on a Qiagen QIAxcel. Equimolar pooling of libraries was performed to generate a final library pool at 4 nM. The pooled library was denatured and diluted according to Illumina protocols and loaded at 8 pM with 10% PhiX control (Illumina) for sequencing. Sequencing was performed on an Illumina MiSeq platform using a MiSeq Nano v2 reagent kit (2 × 150 bp paired-end reads). Demultiplexing and generation of FASTQ files were carried out using Illumina's MiSeq Reporter software. FASTQ files further analysed on CRISPResso2 software for successful insertion[41].

### HiBiT degradation assays and high-throughput screening (HTS)
Crude compounds were assumed to be 10 mM concentration in DMSO and after generating libraries directly used in biological tests in degradation screens. Stock solutions were serially diluted into white bottom opaque 384-well plates (Perkin Elmer) using Echo500 (Labcyte) and DMSO was backfilled to maintain same DMSO content in each well for all concentrations. GSK3β-HiBiT KI HEK293 cells were seeded onto compounds at density of $5 \times 10^3$ cells in $50 \mu L$ of cell suspension per well using Multidrop™ Combi reagent dispenser (Thermo Fisher Scientific) and centrifuged at RT, $100 \times g$ for 1 min and incubated for 24 h in incubators at $37 \degree C$. HiBiT lytic assay buffer (Promega) prepared according to the manufacturer's instructions and added onto each well using Multidrop™ Combi reagent dispenser. Luminescence signal was measured on PheraStar plate reader (BMG) and obtained data normalised to DMSO conditions using Excel and further analysed using GraphPad Prism (v10.1.2) and degradation curves were generated using 4-parameter non-linear regression fit to calculate $DC_{50}$ and $D_{max}$ values. Data points associated with concentrations displaying a hook effect were excluded prior to $DC_{50}$ and $D_{max}$ calculation.

### Degradation assays and western blotting
HEK293, HAP1, HAP1 CRBN KO and SH-SY5Y cells were plated in either 6 cm plates or 6-well plates at varying densities ($0.5–0.7 \times 10^6$ cells per plate or wells), and MEF cells were plated in 10 cm plates ($1–1.2 \times 10^6$ cells per plate) depending on experimental set up in respective culturing condition as described above. Stock solutions of compounds were prepared as 10 mM in DMSO (Sigma−Aldrich). Working dilutions were made up fresh in DMSO and culturing media or differentiation media was refreshed prior to any compound treatments. For mode of action studies, cells were pre-treated with inhibitors for 2 h: $1 \mu M$ MG132 (Merck, 474790-5 mg), $1 \mu M$ MLN4924 (Merck, 5054770001) or $1 \mu M$ GSK3 inhibitor (compound **29**), before treating with PROTACs for

2, 4 or 24 h as indicated in figure legends. Cells were washed twice with ice-cold PBS (Gibco), harvested in PBS and pelleted, cells were resuspended in RIPA buffer supplemented with complete EDTA-free protease inhibitor cocktail (Roche,11873580001, 1/10 mL) and phosSTOP (Roche, 04906845001, 1 tablet/10 mL) for 20 min lysis on ice then cleared via centrifugation at 16,000 × $g$, 4 °C for 20 min. SH-SY5Y cells were lysed in 500 µL ice-cold lysis buffer (25 mM Tris-HCl (pH 7.4), 50 mM NaF, 100 mM NaCl, 1 mM EDTA, 4 mM EGTA, 1% (v/v) Triton-X-100, 0.09% (w/v) sucrose, 10 mM sodium pyrophosphate, 10 mM sodium vanodate, 0.1% (v/v) β-mercaptoethanol, supplemented with complete mini protease inhibitor tablet (Sigma; 1 tablet/ 10 mL buffer)). Lysates were flash frozen, thawed and centrifugated at 16,000 × $g$, 4 °C for 20 min. Following clearance of lysates, protein concentrations were determined by the BCA assay (Fisher Scientific, 23225) or Bradford assay (Bio-rad). Five to thirty micrograms of protein from lysates were prepared in 4×LDS buffer (Thermo Fisher) and 50 mM dithiothreitol (DTT) denatured at 95 °C for 5 min. Proteins were separated on NuPAGE 4−12% Bis-Tris gels (Invitrogen) or 4−12% Novex Tris-glycine gels (Invitrogen) in MOPS buffer (Invitrogen) and transferred onto nitrocellulose membranes using an iBlot3 system (Thermo Fisher Scientific) or nitrocellulose membranes (Cytiva) in cold tris-glycine-methanol buffer (25 mM Tris (pH 7.5), 192 mM glycine, 20% (v/v) methanol). Membranes were blocked with either 5% fat-free milk (Marvel) or 1% (w/v) BSA (Sigma) in TBS-T (20 mM Tris (pH 7.5), 150 mM NaCl, 0.1% (v/v) Tween-20 (Sigma)) at room temperature for 30 min, or in the case of SH-SY5Y assays, 1 h, before incubating with primary antibodies overnight at 4 °C on a rocker. The following primary antibodies were used as indicated dilutions in 1% (w/v) or 5%(w/v) BSA (Sigma) in TBS-T: Anti-GSK3β (Abcam, ab93926, 1:1000), anti-phospho Ser9 GSK3β (CST, 5558 T, 1:1000), anti-phospho Tyr216 GSK3β (Proteintech, 29125-1-AP, 1:1000), anti-GSK3α (Abcam, ab40870, 1:1000), anti-GSK3α + β (Abcam, ab185141, 1:1000), anti-Cul2 (Abcam, ab166917, 1:1000), anti-UB (P4D1) (Enzo Life Sciences, BML-PW093001, 1:1000), anti-HIBIT (Promega, N7200 1:500), anti-c-Myc (Cell Signalling Technologies, 5605S, 1:1000), anti-GSK3 (Cell Signalling Technologies, 5676, lot 7, 1:1000), anti-β-actin (Sigma-Aldrich, A3853, 1:2000), anti-phospho (Thr 509/514) CRMP2 (MRC-PPU Reagents and Services, 1:1000), anti-β-catenin (Cell Signalling Technologies, 9562, 1:1000). Membranes were washed three times for 5 min at room temperature with TBS-T on shaker and incubated with either fluorescently labelled or HRP-linked secondary antibodies for 1 h at room temperature: anti-mouse IRDye 680RD (LiCOR, 926-68070, 1:5,000), anti-rabbit IRDye 800CW (LiCOR, 926-32211, 1:5,000), anti-mouse StarBright blue 520 (Bio-Rad, 12005866, Bio-Rad, 1:5,000), anti-Sheep IgG (H + L) Cross-Adsorbed Secondary Antibody, DyLight™ 800 (Thermo Fisheer Scientific, SA5-10060, 1:5000), anti-rabbit (LiCOR, 926-32211, 1:10,000), anti-mouse (Invitrogen, A21057, 1:10,000), HRP-linked anti-sheep (Invitrogen, 31480, 1:5000), HRP-liked goat anti-rabbit (Invitrogen, 31460, 1:5000). For loading control, hFAB™ Rhodamine Anti-GAPDH Primary Antibody (Bio-Rad, 12004167, 1:10,000), hFAB Rhodamine Anti-Actin Primary Antibody (Bio-Rad, 12004164, 1:10,000) or Anti-actin (Monoclonal Antibody (BA3R), DyLight™, 1:10,000) were used. Images were either captured by using Chemi-Doc™ MP Imaging System (Bio-Rad, fluorescence and chemiluminescence) and band intensities were quantified using Image Lab software (Bio-Rad, v6.1) or captured by LiCOR Odyssey (fluorescence) or Invitrogen iBRIGHT 1500 (chemiluminescence) and quantified using LiCOR imageStudio software (v95.2 or v6.0). For an example of presentation of full blot scans, see the Source Data file.

## Tissue extraction and preparation
A fraction of frozen brain and liver samples from control (CNT) and compound KH1 (**24**) treated animals were transferred into Precellys 2 mL ceramic bead tubes containing 1 mL SDS lysis buffer (50 mM Tris-HCl pH 7.4, 150 mM NaCl, 1 mM EGTA, 270 mM sucrose, 1% (v/v) Triton-X100) supplemented with complete EDTA-free protease inhibitor cocktail (Roche) and PhosSTOP phosphatase inhibitor tablets (Roche) and placed immediately on ice. Then samples were placed in Precellys Evolution tissue homogeniser (Bertin Technologies) and tissues were lysed at 0 °C for 3 × 30 s of cycles at 6800 rpm, with 30 s pause in between cycles. Samples were centrifuged at 4 °C, 2000 × $g$ for 60 s to get rid of bubbles and supernatant was transferred into new tubes. Lysates were cleared via centrifugation at 16,000 × $g$, 4 °C for 20 min. Protein concentration was determined by BCA assay (Thermo Scientific) and immunoblotting performed as previously described above. Immunoblots were quantified using ImageLab (Bio-Rad) and band intensities were normalised to loading controls; anti-GAPDH or Ponceau S staining and fold changes for protein of interest were plotted.

## Multiplexing degradation and cell viability assays
For viability tests, GSK3β-HiBiT KI HEK293 cells were seeded onto white-opaque 96-well plates at a density of $2 \times 10^4$ cells per well. After 16 h, cells were treated with test compounds or DMSO with indicated concentrations for 2, 4 and 24 h. Thirty minutes before the end of treatment, 20 µL of reaction buffer containing Gly-Phe-A FC substrate (1:200 in cell culture media, Chem Cruz, sc506265) added to the each well and cells were returned to cell culture incubator. At the end of the treatment time, HiBiT lytic reagents were prepared according to the manufacturer's instructions (Promega) and 120 µL of reagent dispensed to each well. Fluorescence (viability) and luminescence (degradation) signals were measured by GloMAX discover plate reader (Promega) at the same time. Obtained data analysed using Excel and GraphPad Prism (v10.1.2) as previously described above.

## Kinetic live cell degradation assays
For kinetic degradation assays, GSK3β-HiBiT KI HEK293 cells were reverse transfected into white bottom opaque 96-well assay plates. Briefly, GSK3β-HiBiT KI HEK293 cell suspension at a density of $2 \times 10^6$ per mL was prepared in standard culturing media. 10 mL of cell suspension mixed with transfection mix (1 mL of OptiMEM (Gibco) containing 3:1 ratio of FuGENE HD (Promega) and LgBit (Promega)). Then, cells were seeded into 96-well assay plate as a 100 µL cell suspension per well. After 24 h of transfection, the cell media was removed and 90 µL of $CO_2$-independent medium supplemented with 10% FBS (Gibco) and 1:100 dilution of Endurazine (Promega) and incubated for 2.5 h at 37 °C. Test compounds were added onto cells and luminescence signals were monitored and recorded on GloMAX discover plate reader (Promega) in kinetic mode every 5 min. Data were normalised to DMSO conditions and plotted over time and degradation rates were calculated via Michaelis–Menten non-linear fitting analysis using GraphPad Prism (v10.1.2).

## Serum shift assay
Test compounds were serially diluted into white bottom opaque tissue culture grade 384-well assay plates (Perkin Elmer) using Echo500 (Labcyte) and GSK3β-HiBiT KI HEK293 cells were seeded at a density of $5 \times 10^3$ cells per well either in 50 µL of DMEM supplemented with regular 10% FBS (Gibco) or instead 10% mouse serum (Balb/C) and plates were incubated for compound treatment in incubators at 37 °C. After 24 h of compound treatment, HiBiT lytic reagents (Promega) prepared according to manufacturers' instructions were added into plates and luminescence signal was measured on a PheraStar plate reader (BMG) and data analysed using GraphPad Prism (v10.1.2) as previously described above.

## Quantitative proteomics
For unbiased degradation profiling of compounds at the proteome level, HEK293 cells were seeded onto 10 cm plates at a density of $2-2.5 \times 10^6$ cells per plate. When cells were confluent, culturing media was refreshed and treated with 10 nM final concentration of test

compounds KH1 (**24**) and KH2 (**26**) and non-degrading negative control compounds (**27** and **28**) and DMSO as vehicle control for 2 h in $n = 3$ biological replicates. After treatment, cells were washed twice with PBS (Gibco), collected in Lobinding tubes (Merck) and lysed in RIPA buffer supplemented with Protease (Roche) and Phosphatase (Roche) inhibitors as previously described for 20 min on ice. Lysates were cleared via centrifugation and protein concentrations were determined by using Pierce BCA assay (Thermo Scientific). Similarly, brain and liver tissues from control and KH1 (**24**) treated mice were lysed as described above. One hundred and fifty micrograms of protein from tissues or cells for each condition was aliquoted into LoBind tubes (Eppendorf) and the STRAP (Protifi) protocol was followed according to manufacturers' instructions with slight modifications. Briefly, liquid protein samples mixed with 2× Lysis buffer (10%SDS, 100 mM TEAB pH 8.5), protein was reduced with 5 mM DTT final concentration at 55 °C for 15 min, alkylated (final concentration 20 mM IAA or CAA) at room temperature for 10 min and then acidified (12% phosphoric acid). Protein was then trapped into S-trap columns and washed with S-trap binding/wash buffer (100 mM TEAB in 90% methanol) via centrifugation at $4000 \times g$ for 30 s. Cleaned proteins then went through tryptic digestion (Trypsin (Thermo Scientific) in 50 mM TEAB) overnight at 37 °C or 2 h at 47 °C. Digested peptides were sequentially eluted in 80 μL of 50 mM TEAB (Sigma–Aldrich), 0.2% formic acid (Sigma–Aldrich) and 50% acetonitrile (Sigma–Aldrich). Elutions were pooled and dried down. Dried and cleaned peptides were reconstituted in 110 μL of 100 mM TEAB. TMT-16-plex reagent (Thermo Scientific) was brought into room temperature and reconstituted in 40 μL of anhydrous acetonitrile. TMT reagent solution was transferred to the peptides and incubated on thermoshaker (Eppendorf) at 400 rpm at room temperature for 1 h in the dark. Peptides were quenched by the addition of 8 μL of 5% Hydroxylamine for 1 h at room temperature. TMT-labelled peptides were pooled together and dried down in the speedVac for downstream processing. Pooled peptides were separated by basic reverse phase chromatography fractionation on a $C_{18}$, column with flow rate at 200 μL/min with two buffers: buffer A (10 mM ammonium formate, pH 10) and buffer B (80% ACN, 10 mM ammonium formate, pH 10). Peptides were resuspended in 100 μL of buffer A (10 mM ammonium formate, pH 10) and resolved on a $C_{18}$ reverse phase column by applying a non-linear gradient of 7–40%. A total of 80 fractions were collected and concentrated into 30 fractions.

The dried peptides were reconstituted in 0.1% formic acid and analyzed on an Orbitrap Ascend mass spectrometer coupled to a Thermo Fisher Scientific Vanquish Neo UHPLC. The peptides were separated on an analytical column (Acclaim PepMap RSLC C18, 75 μm × 50 cm, 2 μm, 100 Å) at a flow rate of 300 nL/min, using a step gradient of 2–7% solvent B (90% ACN/0.1% FA) for the first 6 min, followed by 7–18% up to 89 min, 18–27% up to 89–114 min and 27–35% to 114–134 min. The total run time was set to 155 min. The mass spectrometer was operated in a data-dependent acquisition mode in SPS MS3 (FT-IT-HCD-FT-HCD) method. A survey full scan MS (from m/z 350–1500) was acquired in the Orbitrap at a resolution of 120,000 at 200 m/z. The AGC target for MS1 was set as $4 \times 105$ and the ion filling time as 50 ms. The precursor ions for MS2 were isolated using a Quadrupole mass filter at a 0.7 Da isolation width, fragmented using a normalised 30% HCD of ion routing multipole and analyzed using ion trap. The top 10 MS2 fragment ions in a subsequent scan were isolated and fragmented using HCD at a 55% normalised collision energy and analyzed using an Orbitrap mass analyser at a 60,000 resolution, in the scan range of 100–500 m/z.

The proteomics raw data were searched using SEQUEST HT search engines with Proteome Discoverer 3.0 (Thermo Fisher Scientific). The following parameters were used for searches: Precursor mass tolerance 10 ppm, fragment mass tolerance 0.1, enzyme: trypsin, Mis-cleavage: −2, fixed modification: carbamidomethylation of

cysteine residues and TMT of lysine and N-terminal, dynamic modification: oxidation of methionine. The data were filtered for 1% PSM, peptide and protein level FDR. A two-tailed $t$-test with 250 randomisations was performed in Perseus software (v1.6.15.0) to assess statistical significance between groups. The resulting data were visualised as a volcano plot using GraphPad Prism (v10.1.2). Only unique peptides were selected for the quantification. A list of identified unique peptides for GSK3α and GSK3β is included in the Source data file provided.

**PhosphoProteomics**

IMAC beads were prepared from Ni-NTA (nitrilotriacetic acid) superflow agarose beads. The nickel was stripped with 100 mM EDTA and incubated in an aqueous solution of 10 mM iron (III) chloride ($FeCl_3$). Dried peptide fractions were reconstituted to a concentration of 0.5 μg/μL in 80% ACN/0.1% TFA. Peptide mixtures were enriched for phosphorylated peptides with 5 μL IMAC beads for 30 min with end-to-end rotation.

Enriched IMAC beads were loaded on Empore C18 silica-packed stage tips. Stage tips were equilibrated with acetonitrile followed by 50% ACN/0.1% FA then 1% FA. The beads with enriched peptides were loaded onto C18 stage tips and washed with 80% ACN/0.1% TFA. Phosphorylated peptides were eluted from IMAC beads with 500 mM dibasic sodium phosphate, pH 7.0. These peptides were washed with 1% FA before elution using 50% acetonitrile in 0.1% FA. The peptides were then dried by SpeedVac and stored at −20 °C until mass spectrometry analysis. The dried phosphopeptides were reconstituted in 0.1% formic acid and analyzed on an Orbitrap Ascend mass spectrometer coupled to a Thermo Fisher Scientific Vanquish Neo UHPLC. The peptides were separated on an analytical column (Acclaim PepMap RSLC C18, 75 μm × 50 cm, 2 μm, 100 Å) at a flow rate of 300 nL/min, using a step gradient of 2–7% solvent B (90% ACN/0.1% FA) for the first 6 min, followed by 7–18% up to 89 min, 18–27% up to 89–114 min and 27–35% to 114–134 min. The total run time was set to 155 min. A survey full scan MS (from m/z 350–1500) was acquired in the Orbitrap at a resolution of 120,000 at 200 m/z. The most intense ions with charge state ≥2 were isolated and fragmented using higher collision dissociation (HCD) fragmentation, with 30% normalised collision energy, and detected at a mass resolution of 50,000 at 200 m/z. The isolation window was set at 1.6. The phosphoproteomics raw data were searched using SEQUEST HT search engines with Proteome Discoverer 3.0 (Thermo Fisher Scientific). The following parameters were used for searches: Precursor mass tolerance 10 ppm, fragment mass tolerance 0.02, enzyme: trypsin, Mis-cleavage: −2, fixed modification: carbamidomethylation of cysteine residues and TMT of lysine and N-terminal, dynamic modification: oxidation of methionine and phosphorylation of serine, threonine and tyrosine. The data were filtered for 1% PSM and peptide level. A two-tailed $t$-test with 250 randomisations was performed in Perseus software (v1.6.15.0) to assess statistical significance between groups. The resulting data were visualised as a volcano plot using GraphPad Prism (v10.1.2).

**Intrinsic clearance (Cli) experiments**

Test compound (0.5 μM) was incubated with female CD1 mouse, male rat Han Wistar and pooled human liver microsomes (Xenotech™; 0.5 mg/mL 50 mM potassium phosphate buffer, pH7.4) and the reaction started with addition of excess NADPH (8 mg/mL 50 mM potassium phosphate buffer, pH7.4). Immediately, at time zero, then at 3, 6, 9, 15 and 30 min an aliquot (50 μL) of the incubation mixture was removed and mixed with acetonitrile (100 uL) to stop the reaction. An internal standard was added to all samples, the samples centrifuged to sediment precipitated protein and the plates then sealed prior to UPLC-MSMS analysis (Xevo TQ-S Micro, Waters™). XLfit (IDBS, UK) was used to calculate the exponential decay and consequently the rate constant ($k$) from the ratio of peak area of test compound to internal standard at each timepoint. The rate of intrinsic clearance (CLi) of each

test compound was then calculated using the following calculation:

$$CLi \, (mL/min/g \, liver) = k \times V \times Microsomal \, protein \, yield \qquad (1)$$

Where $V$ (mL/mg protein) is the incubation volume/mg protein added and microsomal protein yield is taken as 48 mg, 46 mg and 40 mg protein/g liver for mouse, rat and human, respectively. Verapamil (0.5 μM) was used as a positive control to confirm acceptable assay performance.

## RealSOL solubility

Solubility was assessed using an 'in-house' developed method known as 'RealSOL'. This solubility method measures the solubility in physiological strength phosphate buffered saline starting from 10 mM DMSO solutions of the test compounds. Although a 'kinetic'[1] solubility method, the long shaking times appears to give solubility values that reflect more closely those made from solid samples when compared to a previous nephelometric method (unpublished in-house data).

Test compounds were dissolved in DMSO to give 10 mM solutions. Solubility test samples were prepared by adding a volume (5 μL) of the 10 mM solution to a volume (195 μL) of phosphate buffered saline, pH 7.4 (Sigma-Aldrich, Cat no. P4417, made as per manufacturer's instructions). This solution was then mixed for 24 h (rotary mixing, 900 rpm, 25 °C) excluding light.

After mixing, the solubility test samples were filtered to remove any undissolved material using a proprietary filter (Millipore Multiscreen HTS filter, 96-well format). Samples were drawn through the filter using a vacuum.

The filtrate from the above was analysed for dissolved drug compound using a truncated UHPLC methodology. A Shimadzu Nexera X2 UHPLC system was used, with a reversed-phase column and a simple formic acid gradient elution. The UHPLC parameters are shown in Supporting Information Table 5.

A calibration solution was prepared in the following way: the same 10 mM solution used to prepare the solubility test sample was diluted in DMSO to give a 500 μM solution. This solution was then again diluted with 50:50 acetonitrile: water to give a 50 μM solution. Aliquots (0.2, 2.0 and 5.0 μL) of this 50 μM solution were then injected onto the UHPLC system and the areas of the resultant peaks integrated to produce a calibration line. Aliquots of the test sample filtrate (0.4 and 5.0 μL) were then injected onto the UHPLC system and the resultant peak areas for any peaks corresponding to the test compound determined and quantified using the calibration line (the injection volume that gave a peak area closest to the calibrated range was used for determining solubility).

## Plasma protein binding experiments

A 96-well equilibrium dialysis apparatus was used to determine the free fraction in plasma for each compound (HT Dialysis LLC, Gales Ferry, CT). Membranes (12–14 kDA cut-off) were conditioned in deionised water for 60 min, followed by conditioning in 80:20 deionised water: ethanol for 20 min, and then rinsed in isotonic buffer before use. Female CD1 mouse, Male rat Han Wistar and human plasma was removed from the freezer and allowed to thaw on the day of experiment. Thawed plasma was then centrifuged (Allegra X12-R, Beckman Coulter, USA), spiked with test compound (final concentration 10 μg/mL), and 150 μL aliquots (n = 6 replicate determinations) loaded into the 96-well equilibrium dialysis plate. Dialysis vs isotonic buffer (150 μL) was carried out for 5 h in a temperature-controlled incubator at ca. 37 °C (Barworld Scientific Ltd, UK) using an orbital microplate shaker at 100 revolutions/minute (Barworld Scientific Ltd, UK). At the end of the incubation period, 50 μL aliquots of plasma or buffer were transferred to a 96-well 2.1 mL Waters ™ plate and the composition in each well balanced with control fluid (50 μL), such that the volume of buffer to plasma is the same. Sample extraction was performed by the

addition of 200 μL of acetonitrile containing an appropriate internal standard. Samples were allowed to mix for 1 min and then centrifuged at 3000 rpm in 96-well blocks for 15 min (Allegra X12-R, Beckman Coulter, USA), after which 150 μL of supernatant was removed to 50 μL of water. All samples were analysed by UPLC-MS/MS. The unbound fraction was determined as the ratio of the peak area in buffer to that in plasma.

## Brain binding experiments

As for plasma protein binding experiments, 96-well equilibrium dialysis apparatus was used to determine the free fraction in brain tissue homogenate for each compound (HT Dialysis LLC, Gales Ferry, CT). Membranes (12–14 kDA cut-off) were conditioned in deionised water for 60 min, followed by conditioning in 80:20 deionised water: ethanol for 20 min, and then rinsed in isotonic buffer before use. Frozen brain tissue was thawed and homogenised in artificial CSF at a ratio of 3:1. Homogenate was then spiked with test compound (final concentration 10 μg/mL), and 150 μL aliquots (n = 6 replicate determinations) loaded into the 96-well equilibrium dialysis plate. Dialysis vs artificial CSF was carried out for 5 h in a temperature-controlled incubator at ca. 37 °C (Barworld Scientific Ltd, UK) using an orbital microplate shaker at 100 revolutions/minute (Barworld Scientific Ltd, UK). 150 μL aliquots of spiked homogenate were stored at 37 °C and 4 °C for the course of the experiment to determine recovery. At the end of the incubation period, 50 μL aliquots of homogenate or buffer were transferred to a 96 deepwell plate (Waters™) and the composition in each was balanced with control fluid (50 μL), such that the volume of buffer to brain homogenate is the same. Sample extraction was performed by the addition of 200 μL of acetonitrile containing an appropriate internal standard. Samples were allowed to mix for 1 min and then centrifuged at 3000 rpm in 96-well blocks for 15 min (Allegra X12-R, Beckman Coulter, USA), after which 150 μL of supernatant was removed to 50 μL of water. All samples were analysed by UPLC-MS/MS. The unbound fraction was determined as the ratio of the peak area in buffer to that in homogenate using the following equation:

$$Afu = \frac{Buffer}{Matrix} \qquad (2)$$

where $Afu$ means apparent fraction unbound
Buffer = Analyte/IS ratio in buffer side
Matrix = Analyte/IS ratio in matrix side

$$fucr = \frac{\frac{1}{D}}{\left[ \left( \frac{1}{Afu} - 1 \right) + \frac{1}{D} \right]} \qquad (3)$$

Where $fucr$ means fraction unbound corrected
$D$ is the dilution factor

$$TB = (1 \, fucr) \cdot 100 \qquad (4)$$

where TB means tissue binding

The recovery of the dialysed samples was calculated by comparing the sum of the peak areas (buffer and matrix) of any test compound at 37 °C to that of the same compound at 4 °C. This is then multiplied by 100% to get the answer in percentage.

## In vivo pharmacokinetics and pharmacodynamics

All experiments were carried out under the authority of a licence granted by the UK Home Office under the Animals (Scientific Procedures) Act 1986. Prior to submission to the Home Office, the application for the project licence was approved (approval number WEC2024−19) by the University Welfare and Ethical Use of Animals

Committee, acting in its capacity as an Animal Welfare and Ethical Review Body as required under the Act.

Fifteen female mice of the Balb/c strain were obtained from Charles River Laboratories, UK. Animals were maintained under a 12-h light/12-h dark cycle, in Thoren mouse caging containing EC0 aspen bedding (DBM, UK), with 'Sizzlenest' and 'Rodent Rolls' nesting material (DBM). Enrichment materials included aspen chew sticks, cardboard houses, tunnels, swings, and sunflower seeds. All animals had *ad libitum* access to food (5LF2 Extruded Diet, IPS Ltd, UK) and tap water throughout. Temperature and relative humidity were maintained between 19 °C–24 °C and 45%–65%, respectively. All animals received a minimum of 10 days of acclimatisation prior to the start of the study.

**Cassette PK study with 22, 25 and 26.** The cassette formulation (**22, 25** and **26**) was prepared by producing separate stock solutions of each compound dissolved in DMSO (dimethyl sulfoxide) at a concentration of 6 mg/mL. Aliquots of each stock solution were combined in equal parts to produce a stock solution containing 2 mg/mL of each compound. This solution was then diluted with Polyethylene glycol 400 (PEG400), followed by drop-wise addition of sterile water. The final formulation consisted of 5% DMSO, 40% PEG400, and 55% water, and contained 0.1 mg/mL of each compound. For the cassette PK study, three female Balb/c mice were dosed via a bolus intravenous injection at a dose level of 0.5 mg/kg per compound (5 mL/kg). Serial blood samples (10 μL per sample) were collected from the lateral tail vein of each mouse at 5, 15 and 30 min, 1, 2, 4, 6, 8, and 24 h after dose administration. All blood samples were diluted into 90 μL of Milli-Q ultrapure water and stored at −20 °C prior to bioanalysis. The concentration of **22, 25** and **26** in blood and brain was determined by UPLC-MS/MS.

**Discrete i.v./p.o. KH1 (24) PK study and brain penetration.** The intravenous formulations for KH1 (**24**) were prepared by dissolving the compound in DMSO. This was then diluted with PEG400, followed by dropwise addition of sterile water to form a complete solution. The resulting formulations consisted of 5% DMSO, 40% PEG400, and 55% water, and contained KH1 (**24**) at a concentration of 0.1 mg/mL (for the PK study), or 1.0 mg/mL (for the PK/PD study), respectively.

The oral formulation for KH1 (**24**) was produced by vigorously mixing the compound with a sterile solution containing 0.5% hydroxypropylmethylcellulose in water, until a fine homogenous suspension was formed. The resulting formulation contained KH1 (**24**) at a concentration of 0.6 mg/mL. Dose formulations were prepared on the day of dosing, and animals were observed regularly after dose administration.

For the KH1 (**24**) pharmacokinetic study, six female Balb/c mice were dosed via a bolus intravenous injection at a dose level of 0.5 mg/kg (5 mL/kg), and three female Balb/c mice were dosed via oral gavage at a dose level of 3.0 mg/kg (5 mL/kg). Serial blood samples (10 μL per sample) were collected from the lateral tail vein of three intravenously-dosed mice and three orally-dosed mice at 5, 15 and 30 min, 1, 2, 4, 6, 8, and 24 h after dose administration. Blood samples were diluted into 90 μL of Milli-Q ultrapure water. The remaining three intravenously-dosed animals were placed under terminal anaesthesia with isoflurane, 2 h after bolus administration. Blood was collected via cardiac puncture and the brains were removed. Two aliquots of 10 μL blood were each diluted into 90 μL of Milli-Q ultrapure water. All blood samples were stored at −20 °C prior to bioanalysis. The concentration of KH1 (**24**) in blood and brain was determined by UPLC-MS/MS.

**KH1 (24) intravenous PK/PD study.** The intravenous formulation for KH1 (**24**) was prepared by dissolving the compound in DMSO. This was then diluted with PEG400, followed by dropwise addition of sterile water to form a complete solution. The resulting formulation consisted of 5% DMSO, 40% PEG400, and 55% water, and contained KH1 (**24**) at a concentration of 1.0 mg/mL.

For the PK/PD study for KH1 (**24**), three female Balb/c mice were given a bolus intravenous injection at a dose level of 5 mg/kg (5 mL/kg). Animals were observed regularly after dose administration. At 4 h after the administration of KH1 (**24**), all animals were placed under terminal anaesthesia with isoflurane. Blood was collected via cardiac puncture and the brain and liver were removed. Two aliquots of 10 μL blood were each diluted into 90 μL of Milli-Q ultrapure water. All diluted blood samples were stored at −20 °C prior to bioanalysis. A portion of the brain was stored at −20 °C awaiting bioanalysis, while the remaining brain and liver tissues were snap-frozen in liquid nitrogen and stored at −80 °C awaiting biological investigation. The concentration of KH1 (**24**) in whole blood and brain tissue was determined via UPLC-MS/MS.

**MS blood analysis.** Blood samples (10 μL) from study mice were diluted 1× with 90 μL of water immediately when the sample was taken. All calibration, quality control, blanks and double blank samples (all 100 μL volume) were mixed (vortexed) with 300 μL of LCMS grade acetonitrile to precipitate proteins.

For study samples, a volume (30 μL) was mixed (vortexed) with a volume (90 μL) of acetonitrile to precipitate proteins. After protein precipitation, all samples were centrifuged to pellet the proteins. Study samples in Micronics tubes were centrifuged using a Beckman Coutler Allegra C-12R Centrifuge for 10 min at 3750 rpm. All other samples in Eppendorf tubes were centrifuged using a Sigma SciQuip 1–14 centrifuge for 10 min (14,800 rpm/16,163 × $g$). Supernatant from study samples (80 μL) and calibration samples, QCs, blanks, and double blanks (200 μL) were then transferred to HPLC glass vials containing Milli-Q water (40 μL or 100 μL, respectively), ready for LCMS analysis.

### Statistical analysis

Data are represented with mean averages plotted, ± Standard Deviation (SD). Statistical analyses were performed using GraphPad Prism Software (v10.1.2). Unpaired Two-tailed Student's $t$-tests were used to compare treatments from the in vivo pharmacodynamic study. $DC_{50}$ and $D_{max}$ values were calculated by using 4-parameter non-linear regression curve fit by excluding hooking points $DC_{50}$ and $D_{max}$ calculation. Degradation rates were calculated by Michaelis−Menten non-linear fitting analysis. Significance threshold of $p < 0.05$ (two-tailed) was considered for all tests. Sample sizes for each experiment are indicated in the respective figure legends.

### Ethics statement

All studies described comply with all relevant ethical regulations and were carried out under the authority of a licence granted by the UK Home Office under the Animals (Scientific Procedures) Act 1986. Prior to submission to the Home Office, the application for the project licence was approved (approval number WEC2024-19) by the University Welfare and Ethical Use of Animals Committee, acting in its capacity as an Animal Welfare and Ethical Review Body as required under the Act.

### Reporting summary

Further information on research design is available in the Nature Portfolio Reporting Summary linked to this article.

## Data availability

All processed and raw data generated in this study are provided in the Source Data, Supplementary Information (for Supplementary Figs. only) and Supplementary Data files. All mass spectrometry proteomics data for KH1 (**24**) has been deposited in the PRIDE data repository under accession codes: PXD065662 (24 h degrader treatment, HEK293 proteome), PXD065692 (24 h degrader treatment, HEK293

phosphoproteome), PXD065750 (4 h degrader treatment, HEK293 proteome), PXD065724 (4 h degrader treatment, HEK293 phospho-proteome), PXD065760 (2 h degrader treatment, HEK293 proteome), PXD065722 (2 h degrader treatment, HEK293 phosphoproteome), PXD065740 (mouse liver proteome following degrader treatment), and PXD065731 (mouse liver phosphoproteome following degrader treatment). Data previously published under PDB accession codes 5K5N and 8DJC were used to inform GSK3 binder design.

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

## Acknowledgements

This project received funding from UKRI Biotechnology and Biological Sciences Research Council (BBSRC) grant number BB/T00875X/1 (A.H.), UK Engineering and Physical Sciences Research Council (EPSRC) grant number EP/X020088/1 (W.F., N.M.K.) and Biotechne (Tocris) (A.H., W.F.) and by the Wellcome Trust Award 226943/Z/23/Z (W.F.).Diabetes UK research grant ref:20/0006178 also supported (C.S.). We are thankful to Dr Conner Craigon for guide RNA design and University of Dundee FACs facility for single cell sorting during HiBiT knock-in cell line generation; the Ciulli lab for the gift of the HAP1 WT and CRBN KO cell lines; the University of Dundee National Phenotypic Screening Centre and especially Dr Alistair Langlands for enabling training and access to liquid handling systems; the University of Dundee Proteomics facility for assisting with MS-proteomics data analysis, MRC Reagents and Services and Dario Alessi for enabling access to tissue processing equipment; MRC Reagents and Services for Amplicon sequencing, Lourdes Acosta Benavides for synthesis of PF-367 related building blocks; Lorna Campbell, Hali Joji, Nicole Mutter, Jodie Oates, Yoko Shishikura for support with in vitro ADME studies; the Biological services facility where the animal work was carried out; Erika Pinto and Frederick Simeons for assisting with pharmacokinetics; Laura Frame for sample preparation and bioanalytics; Dr Lisa Logie for her assistance with obtaining data on GSK3 substrates for differentiated SHSY5Y cells and Louise Sargent and Rebecca Craik for helping with access to LCMS facilities.

## Author contributions

A.H. and N.M.K. contributed equally and will be putting their name first on the citation in their curriculum vitae. W.F. conceived the idea, acquired research funds and directed the project. H.M., J.C-B. and G.M. provided input to project design. W.F. and A.H. designed compounds. W.F., A.H. and K.J. performed critical synthetic routes specifically related to the library construction presented herein. N.M.K. generated cell lines, developed high throughput screening assays and performed compound screening, identified and validated hits and profiled chemical probes. W.F., N.M.K. and K.R. designed experiments and interpreted data. J.R. and S.B. supervised and performed in vitro ADME profiling. C.D. supervised and performed in vivo pharmacokinetic and pharmacodynamic studies. C.S. designed and N.M. performed SH-SY5Y related cellular assays and data analysis. N.M.K. and G.S. designed and performed whole and phospho-proteome sample preparation and mass spectrometry experiments, including related data processing and analysis. W.F., N.M.K. and A.H. wrote the manuscript with input from all other authors.

## Competing interests

The authors declare no competing interests.
