## [Transparent Peer Review file · Nature Communications]

Discovery of a CNS active GSK3 degrader using orthogonally reactive linker screening

Corresponding Author: Dr William Farnaby

Version 0:

Reviewer comments:

Reviewer #1

(Remarks to the Author)

Thank you for the submission. I loved reading it – well done: great idea, noteworthy results, careful experiments, careful controls

I have a few suggestions.

Introduction: “Furthermore, there is a lack of chemical probes and allied datasets to act as benchmarks to understand the in vitro and in vivo profiles required to achieve degradation of target proteins in the brain.” This sentence is a bit complicated and confusing to me, and I am not clear if it is necessary. Would you mind adding more detail or telling how your work fixes this?

Several times you mention “chemical probe quality”. It might be worth mentioning what this means to you and reference an appropriate paper or two on the topic (probes and/or protac probes).

You say: “degrader of chemical probe quality has not been qualified”. This is confusing to me. Do you just mean “has not been made” or “has not been disclosed”? I find the language with “qualified” to be a bit unclear.

“without the need to purification via chromatography” – SCX is purification, so I think you should temper the language and be consistent. Later you call it “semi-purification” and I think once you call it purification. Your method is great, and I think having this simple plate based SCX purification strategy is useful and a great piece of the puzzle. No need to pretend it is not purification and consistent language will help. You don’t need to oversell.

“in the absence of additives such as potassium iodide” – this seems a bit peculiar to me to have this in here. Is there a reason you said this. If you want it included, I would just emphasize that you developed conditions that were as simple as possible? The fewer reagents the better for direct to biology.

Do you have any evidence for the “ring opening during SCX” of the spirocycles? You say it is likely a problem. Data that supports this would be useful, otherwise perhaps just leave it out? Or hypothesize that it’s the problem and suggest future experiments to overcome it?

I suggest “...E3 ligase and linker types demonstrate the effectiveness and utility of the proposed orthogonally...” (instead of the word “evidence” because I don’t like evidence as a verb here.

Figure 2: please show the parent GSK3 compounds PF-367 and CMP-47 somewhere. Reason: helps us understand your design better, and in the text, you mention the triazole of PF-367 but it’s not visible anywhere.

Figure 2 panel F can you please label the color scale with “uv purity”?

Figure 3 is a bit confusing to me. Please label “b” with the parent ligand to remind us. Please label “d” with the parent ligand to remind us. Figure 2a and c are clearly with varying concentrations. Are b and d single concentrations? It is not clear to me. Presumably these are the same building block layouts as figure 2. Is this right? For figure 3d there is no row G. Why is this? In the legend for figure 3 you say, “compounds 21, 22, 23, 24, 25, and 26 were selected for resynthesis”. Please tell us which wells these are from 2b and 2d.

In 2k and 2 l will you please label the x-axis?

“Qualification of cellular GSK3 degrader probes” - I don't like this as a heading. How about “Evaluation of GSK3 degrader probes” instead

“Ligase binding incompetent negative control compounds 27 and 28” is unnecessarily complicated language and sounds awkward. Maybe more simply “negative control compound that can no longer bind to cereblon”?

Figure 4 – characterization instead of qualification?

Negative control language too complicated to me

KH1 is a brain active GSK3 degrader section:

You use “wherein”, maybe just “and”?

“showed less hook effect” – I suggest that “shows a diminished hook-effect” is more grammatically sound

Reviewer #2

(Remarks to the Author)

TPD is an important therapeutic modality and the development of methods to synthesize and evaluate degraders is of high interest and value. In the present work the authors describe a high throughput method for synthesizing and screening GSK3 degraders using biorthogonal linkers. After meeting a certain purity and DC50 threshold, “hits” were synthesized and purified using standard organic chemistry methods. Endpoint and kinetic degradation was assessed using a stably transfected GSK3-Hibit cell line, and these results were confirmed with Western Blots of GSK3 and downstream targets and proteomic mass spectrometry assays. The mode of degradation was also confirmed with appropriate controls. Finally, in vivo experiments were performed to assess pharmacokinetic parameters and initial pharmacodynamic properties of the lead compounds.

While the work was thorough and well done, there are some questions and comments that should be addressed

1. It would be useful to comment on the potential differences in the phenotypic response between the inhibition and the degradation of GSK3 and if there are any differences between the selective inhibition and/or degradation of just one of the two paralogues. This will provide a better biological rationale for the development of PROTACs that target GSK3 rather than simple inhibitors. Can more discussion be provided on the focus of GSK3 degradation (over GSK) especially in relation to the in vivo results? Why is one isoform degraded more over the other and what are the implications of this?
2. Can the authors comment about the ease of difficulty of synthesizing degraders or inhibitors that cross the BBB? Mention of the brain penetrant Nurix degrader indicates that this is possible but what design principles need to be considered?
3. In the initial screen of the compound library how are the parameter cut offs set that dictate a “hit”.
4. Can the authors comment on why only short linkers with low polarity were used?
5. -The results from compounds with only 50-60% purity seems like it could lead to false positive or false negative results. Can you comment on the use of that >50% purity cut off point? Could this be why some wells never reached 100% DCmax? Do you think potential hits were missed simply due to their lower synthetic yields and lower reactivity of the substrates?
6. -In addition, you mention in line 162 that you assume 100% conversion and purity when you make your 10 mM stocks, but obviously some compounds are much less pure than this. Does this not skew your resulting DC50s?
7. There was no mention of compound toxicity either in cells or in vivo, which is important for assessing the utility of a new drug-like compound. In addition was any toxicity noted in the in vivo (or in cell-based) studies, especially with the high degradation of GSK3 in the liver? Also in Fig. S2A cells are treated with CHX (cycloheximide) for up to 72h. This will presumably affect cell viability since CHX is toxic and the authors should comment on how this might affect the conclusions drawn from the data with appropriate control experiments.
8. The PDB ID “5K4N” was removed from the distribution of released PDB entries (status Obsolete) on 2018-12-05. Details: THE ENTRY IS OBSOLETE PER AUTHORS REQUEST.”
9. Missing reference for Fig S1C-D. There is some confusion on the nomenclature of protein-ligand interactions. Cation- π interactions refer to a positive charged ligand interacting with aromatic amino acid residues (e.g., Phe, Tyr, or Trp) on the target protein (e.g. 10.1038/nature07768; 10.1021/ar300265y). On the other hand, π -cation interactions, refer to an interaction between the π system of a small-molecule ligand with cationic amino acid residues (e.g., protonated Arg or Lys) in the target protein (e.g. 10.1006/jmbi.2000.4033). In addition, in Fig S1D (8DJC) the hydrogen atoms for the ligand are shown while in Fig S1C there are no hydrogen atoms. Maybe the authors could prepare both images removing hydrogen atoms for the small molecules and highlighting the π -cation (and not as written in the text cation- π) interaction of the triazole with the R141, highlighting distances between amino group and triazole.
10. Fig. S2C: GSK3a should be GSK3 α . Also there is less degradation of GSK3 α compared to GSK3 β with compound 26, at

100nm. This behavior could be driven by the Hook effect. But if so would it be expected to be more marked with just one the two paralogues since this would translate to better selectivity for 26 with GSK3 α . However for compound 26 at 10 nM there is more degradation of GSK3 β . Thus, it would be very beneficial comment about the differences in the degradation of the two paralogues by compound 26 and link this with selectivity.

11. Are there any data for the relative expression of the two paralogues in the brain (mRNA or protein expression)?

12. Is anything known about the rates of synthesis of GSK3 α and GSK3 β since this can control TPD efficiency and selectivity.

13. Fig. S2I: What is the concentration of the PROTACs reported in the graph? Why is it missing traces for compounds 23 and 26? In addition, it might be worth focusing on degradation in the first hour, to have a better insight into the initial velocity

14. P.6 l.180: Is a 20-fold difference moderate? In any case it would be useful to provide an explanation for this marked difference between the two synthetic approaches.

15. Fig 3A and C are not very clear since so many data are included in the graphs. However : Fig 3B and D are much clearer. It is confusing that your panels are not in order.

16. Suggest that all instances of "in vivo" and "in vitro" are italicized

17. There may be a typo in line 54 and 144.

18. In Figure 5, there is * above some panels, but it is not specified what p value this represents.

19. A section is included in your methods for tissue processing for the PD study, but you don't include any details of how blood was worked up for MS analysis. Was whole blood analyzed MS without any processing?

20. Suggest combining Table S5 and S6 for simplicity. Is there a way to also combine Tables S2-S4?

21. It would be beneficial to discuss the serum shift assay in more detail and the importance of the results derived from that assay.

22. In your chemistry SI, "calculated" is consistently and incorrectly spelled as "calcμLated".

23. Chemical formulas should have the numbers subscripted.

24. It would be beneficial to include schemes in the chemistry SI.

Reviewer #3

(Remarks to the Author)

Reviewer #4

(Remarks to the Author)

Reviewer #5

(Remarks to the Author)

This manuscript presents a novel screening strategy for identifying novel E3 ligase-target protein modalities for targeted protein degradation, simultaneously varying linker, E3, and target recruiting ligands. The authors describe development and validation of a CNS-penetrant GSK3 degrader using this orthogonally reactive linker platform for screening. The method provides a framework to test design principles for PROTAC generation via sequential synthesis of bifunctional molecules, thereby diverging from the concept of E3 ligase-linker pre-conjugation and identifying alternative chemical modalities mediating GSK3 attenuation compared to previously published GSK3 degraders. The authors identify potent degrader (compound 24) with sub-nM DC-50, rapid GSK3 degradation kinetics in cells, and bioactivity in mice. Mechanism of action analyses reveal dependence upon GSK3 kinase domain and CRBN for degradation. Deeper insight into proteome-level changes (in addition to GSK3 β attenuation) in mouse liver/brain is required to validate the conclusion that KH1 is selective and tolerated at the presented dose in vivo.

Major comments:

1.) A broader proteomic analysis would provide stronger validation for selectivity of compound 24 as a chemical probe and

elucidate possible off target effects with longer durations of treatment. While the selectivity for compound 24 is strong (figure 4D), this compound is modestly perturbing the proteome at 2h. What happens to the proteome at later timepoints including 4h (as presented in MEFs and mouse treatments), or 24h as presented in SH-SY5Y experiments? There are several proteins upregulated with treatment of compound 24; while only 1.5-fold, this effect may be more pronounced with longer durations of treatment. Unique peptides quantified for GSK3a vs. GSK3b? Presenting the peptide info used for quant would be useful in a supplemental figure.

2.) Does compound 24 impact GSK3 activity prior to its degradation? Chemical inhibition has been shown to increase inhibitory phos. at ser 9. Is auto-phosphorylation responsible for the rapid degradation kinetics (GSK3 inhibitor treatment in figure 4F/G), or does GSK3 inhibitor binding prevent compound 24 from accessing the kinase domain? The figure legend notes GSKi 47, methods state compound 20—they are synonymous but carry consistent nomenclature. A phospho-proteome analysis of the differentiated SH-SY5Y cells (western shown in Fig 4i or in HEK293 cells if more permissible) with corresponding whole proteome would strengthen the conclusion that GSK3 substrates are largely de-phosphorylated (or stabilized in the case of those that are degraded following phosphorylation) following treatment. A short timepoint (prior to robust GSK3 degradation) and both 2-4h/24h could be informative to assess phosphorylation-dependent changes and global protein stability following GSK3 attenuation.

3.) Compound 24 (KH1) leads to significant GSK3b attenuation in mouse liver and moderate reduction in mouse brain 4 hours post treatment. If no additional dose is given, do the mice survive normally? It would be interesting to see a whole proteome analysis for naïve, 4h treated, and 4h treated followed by several hours of recovery to examine how the liver and brain proteome may be remodeled in vivo. This may highlight potential off-target or stress signatures before additional doses/treatment regimens are examined in vivo.

Minor comments:

1.) Comparison of ligand 19 and 20—what is the reason for the increased frequency of potent compounds with the latter? It is apparent from the results, but a line describing why this may be the case would be beneficial to the reader. Supplemental Figure S1C, D are not discussed in the manuscript. The results section reads “For GSK3 targeting ligands, we designed azide containing building blocks based on PF-367 and CMP-47(Fig 2D)”—reference the supplemental data here to demonstrate engagement of the GSK3 kinase domain and clarify for the reader that these are ligands 19, 20 respectively.

2.) GSK3b expression as detected by endogenous antibody is lower in HiBiT tagged cells vs. WT. Are both alleles targeted? Stability following CHX is comparable b/w tagged and endo GSK3b in the 72h experiment (S2b) but baseline expression is lower. Showing the 72h experiment should suffice given GSK3b is a long-lived protein.

3.) Figure S3: panel F should be moved to the left and panels H, I shifted to the right. The figure is hard to follow given the order of panels. Furthermore, labels on panel F are likely reversed—compound 24 should exhibit GSK3 degradation while compound 27 is the control, non-degradative compound. Presenting cell viability data in response to 24h of 10 or 100nM compound 24 treatment would be helpful to understand potential toxicity with longer duration or increased concentration.

4.) Figure legends describing statistical data could be expanded slightly to explain error bars and significance notations.

Version 1:

Reviewer comments:

Reviewer #1

(Remarks to the Author)

Thank you for your revisions. In my estimation you have carefully and thoughtfully responded to all the questions, comments, and suggestions put in front of you by the reviewers. The paper was already a fine contribution and now it's better. Well done.

Reviewer #2

(Remarks to the Author)

The authors have satisfactorily addressed my comments.

Reviewer #4

(Remarks to the Author)

Reviewer #5

(Remarks to the Author)

The authors addressed my major points, and I appreciate their commitment to completing the suggested studies. I support acceptance of this impactful study.

Dear Reviewers,

We are grateful for the constructive feedback and provide here a point-by-point response which we hope adequately addresses their requests and questions. We also provide PRIDE accession codes and reviewer log in tokens for all mass spectrometry proteomics data for compound KH1 (**24**) included in this study.

PRIDE reviewer log in tokens:

Project accession: PXD065760

Token: u2EnYRrycBph

Project accession: PXD065750

Token: PGgn5DIqhXUP

Project accession: PXD065731

Token: nVGkeD8Glt1r

Project accession: PXD065724

Token: SkXxHnxdVJa9

Project accession: PXD065722

Token: mTuxhT6cBMbw

Project accession: PXD065692

Token: NvnngiTsmVqAQ

Project accession: PXD065662

Token: pO2KwoBKAttV

Project accession: PXD065740

Token: tBN6AZp46bHG

REVIEWER COMMENTS

Reviewer #1 (Remarks to the Author):

Thank you for the submission. I loved reading it – well done: great idea, noteworthy results, careful experiments, careful controls

We are delighted that the referee enjoyed reading our work and we are grateful for the proposed changes, which we agree will further strengthen our manuscript. We agree with all the proposals and sought to amend accordingly as detailed below.

I have a few suggestions.

Introduction: “Furthermore, there is a lack of chemical probes and allied datasets to act as benchmarks to understand the in vitro and in vivo profiles required to achieve degradation of target proteins in the brain.” This sentence is a bit complicated and confusing to me, and I am not clear if it is necessary. Would you mind adding more detail or telling how your work fixes this?

We have now simplified this sentence and merged it with the previous sentence, which we hope clarifies and provides more context.

Despite these advances, the property space, design principles and assay cascades required for CNS PROTAC discovery are not established, with a lack of chemical probes and datasets that could guide further understanding.

Several times you mention “chemical probe quality”. It might be worth mentioning what this means to you and reference an appropriate paper or two on the topic (probes and/or protac probes).

We have now cited (line 86, page 2, reference 26) a cross-industry/academia perspective from others in the field who have eloquently captured and defined this and to which our molecule adheres. *J. Med. Chem.* 2023, 66, 14, 9297–9312

You say: “degrader of chemical probe quality has not been qualified”. This is confusing to me. Do you just mean “has not been made” or “has not been disclosed”? I find the language with “qualified” to be a bit unclear.

We have amended this to ‘disclosed’ (page 2 line 86)

“without the need to purification via chromatography” – SCX is purification, so I think you should temper the language and be consistent. Later you call it “semi-purification” and I think once you call it purification. Your method is great, and I think having this simple plate based SCX purification strategy is useful and a great piece of the puzzle. No need to pretend it is not purification and consistent language will help. You don’t need to oversell.

We appreciate there is a lack of consistency here and so have sought to address this as follows:

Top of page 3:

We aimed to achieve this with methods that could be performed in a plate-based setting, with no protecting groups required, and that would result in compounds of sufficient purity that they could be tested directly in cellular screens without the need for purification via chromatography.

Has become...

We aimed to achieve this with methods that could be performed in a plate-based setting, with no protecting groups required, and with a workflow that avoided column chromatography.

“in the absence of additives such as potassium iodide” – this seems a bit peculiar to me to have this in here. Is there a reason you said this. If you want it included, I would just emphasize that you developed conditions that were as simple as possible? The fewer reagents the better for direct to biology.

Many thanks for this suggestion, we have removed specific comments on KI and included a comment on identifying the simplest conditions possible (top of page 4), we now state

Based on these criteria, we identified the simplest conditions possible for both S_N2 and CuAAC “click” chemistries as a first model system (Fig 1B).

Do you have any evidence for the “ring opening during SCX” of the spirocycles? You say it is likely a problem. Data that supports this would be useful, otherwise perhaps just leave it out? Or hypothesize that it’s the problem and suggest future experiments to overcome it?

We have removed the comment speculating on ring-opening as this was observed in only some samples on LCMS and would need further study to prove.

I suggest “...E3 ligase and linker types demonstrate the effectiveness and utility of the proposed orthogonally...” (instead of the word “evidence” because I don’t like evidence as a verb here.

We have changed ‘evidence’ to ‘demonstrate’

Figure 2: please show the parent GSK3 compounds PF-367 and CMP-47 somewhere. Reason: helps us understand your design better, and in the text, you mention the triazole of PF-367 but it’s not visible anywhere.

Many thanks for this comment. We have now made changes to Figure 2 to capture the fact that the ligands shown are modified versions rather than the parent ligands/inhibitors themselves and to make clear where they are used. We have also added the parent inhibitor structures to Supporting Figure S1.

Figure 2 panel F can you please label the color scale with “uv purity”?

We have done this as requested.

Figure 3 is a bit confusing to me. Please label “b” with the parent ligand to remind us. Please label “d” with the parent ligand to remind us. Figure 2a and c are clearly with varying concentrations. Are b and d single concentrations? It is not clear to me. Presumably these are the same building block layouts as figure 2. Is this right? For figure 3d there is no row G. Why is this?

We have now amended figure 3 by adding the GSK3 binder reagents next to the corresponding plates. Figures c and d (formerly b and d) are heatmaps labelled according to the DC_{50} value. Row G was not included in plate 2 due to building block availability when running the screen.

In the legend for figure 3 you say, “compounds 21, 22, 23, 24, 25, and 26 were selected for resynthesis”. Please tell us which wells these are from 2b and 2d.

We have added this detail as proposed in the figure 3 legend.

In 2k and 2 l will you please label the x-axis?

Many thanks for spotting this. We have amended panels 3K and 3L.

“Qualification of cellular GSK3 degrader probes” - I don’t like this as a heading. How about “Evaluation of GSK3 degrader probes” instead

We have amended this as suggested

“Ligase binding incompetent negative control compounds 27 and 28” is unnecessarily complicated language and sounds awkward. Maybe more simply “negative control compound that can no longer bind to cereblon”?

We have changed:

*We evaluated endogenous degradation of both GSK3 α and GSK3 β at 2 and 4 hours in HEK293 cells and compared performance to ligase binding incompetent negative control compounds **27** and **28** (Fig S3D-E).*

To...

*We evaluated endogenous degradation of both GSK3 α and GSK3 β at 2 and 4 hours in HEK293 cells and compared performance to negative control compounds (**27** and **28**) that are unable to engage CRBN, due to methylation of the dihydrouracil ring. (Fig S3D-E).*

Figure 4 – characterization instead of qualification?

This has been amended as proposed by the referee

KH1 is a brain active GSK3 degrader section:
You use “wherein”, maybe just “and”?

This has been amended as proposed by the referee

“showed less hook effect” – I suggest that “shows a diminished hook-effect” is more grammatically sound

This has been amended as proposed by the referee

Reviewer #2 (Remarks to the Author):

TPD is an important therapeutic modality and the development of methods to synthesize and evaluate degraders is of high interest and value. In the present work the authors describe a high throughput method for synthesizing and screening GSK3 degraders using biorthogonal linkers. After meeting a certain purity and DC50 threshold, “hits” were synthesized and purified using standard organic chemistry methods. Endpoint and kinetic degradation was assessed using a stably transfected GSK3-Hibit cell line, and these results were confirmed with Western Blots of GSK3 and downstream targets and proteomic mass spectrometry assays. The mode of degradation was also confirmed with appropriate controls. Finally, in vivo experiments were performed to assess pharmacokinetic parameters and initial pharmacodynamic properties of the lead compounds.

While the work was thorough and well done, there are some questions and comments that should be addressed

1. It would be useful to comment on the potential differences in the phenotypic response between the inhibition and the degradation of GSK3 and if there are any differences between the selective inhibition and/or degradation of just one of the two paralogues. This will provide a better biological rationale for the development of PROTACs that target GSK3 rather than simple inhibitors. Can more discussion be provided on the focus of GSK3 β degradation (over GSKA) especially in relation to the in

vivo results? Why is one isoform degraded more over the other and what are the implications of this?

We would not claim obvious specificity or biasing of KH1 for either paralog based on the presented data set so have not expanded on that here. However, we do acknowledge that selective paralog removal is attractive and pursuit of a more paralog specific GSK3 modifier is the subject of ongoing studies. Furthermore, we found that the antibody for detecting mouse GSKa gave very noisy data in MEF treated samples, making it challenging to assess the GSKa DC50 in cells, prior to in vivo PD experiments. We therefore did analyse GSK3 α in tissues via immunoblot given the noisy data observed with this antibody in cells. However, whilst total GSK3 α could not be quantified in the total liver proteomics data now included (figure 6), we did manage to quantify Ser21 GSK3 α phosphorylation in the phosphoproteomics analysis, supporting that GSK3 α is also degraded in these samples. Therapeutically, there is more understood regarding the potential of GSK3 β , however this does not make GSK3 α any less attractive for investigation in our view.

With respect to the broader question of utility of a degrader vs an inhibitor for GSK3, based on there being limited scaffolding function reported for GSK3 paralogs, we would anticipate broadly similar phenotypes. However, we do hypothesise that there may be differences in opportunities between these two modalities with respect to how the abundance/activity of GSK3 paralogs can be controlled, especially kinetically, to obtain differential downstream impact on the phosphorylation of GSK3 substrates. E.g. it may be that a degrader offers the chance for more control over extent of GSK3 shutdown to achieve safe but effective disease pathway modulation, compared with inhibitors which often have high potency combined with high and long exposure. A degrader also provides the advantage of an obvious biomarker to monitor this (i.e. GSK3 abundance). We believe the probe molecule presented here (KH1) offers opportunities for future work that can investigate this more thoroughly but is not the major focus here. Nevertheless, additional data now provided does, we feel, better illuminate the opportunities. E.g. Interrogating GSK3 signalling in a manner not achievable with inhibitors or genetic knock-down/out due to its specificity, rapid degradation kinetics and ability to remove GSK3 at concentrations that do not achieve levels of occupancy required for pre-degradation inhibition (see figure 4G). This is also exemplified by the phosphoproteomic data acquired at different time points following GSK3 removal, that is now added in figure 5.

In the discussion we have also added

This data highlights the opportunities that are now enabled to further interrogate how rapid removal of GSK3 paralogs may impact the activity, particularly kinetically, of disease relevant substrates as compared with those that may be deemed to be safety risks, and how this compares to inhibitors in more complex disease models.

2. Can the authors comment about the ease of difficulty of synthesizing degraders or inhibitors that cross the BBB? Mention of the brain penetrant Nurix degrader indicates that this is possible but what design principles need to be considered?

Thanks for this question. As compared with classical inhibitors, the principles for designing brain penetrant/active degraders are not established and hence we feel this study provides an important publicly available data set towards understanding such principles. We are encouraged by the disclosures by Nurix and Arvinas regarding their brain penetrant clinical stage PROTACs as it supports that the end goal is achievable. However, the question of how this is achievable, prospectively, remains unanswered. We do not go as far as to suggest we have solved this problem in a single study, of course, but we do feel this work provides the first open data set for interrogation and to

act as a benchmark for the field to start learning what these principles may be. Due to the demands of a concise introduction we have tried to capture this in simple terms as follows (n.b. this is amended compared to the first version based on comments from referee 1):

Despite these advances, as compared with inhibitors⁵, the property space, design principles and assay cascades required for CNS PROTAC discovery are not established, with a lack of chemical probes and datasets that could guide further understanding.

N.B. reference 5, now added, is a seminal report by medicinal chemists at pfizer on their use of multi-parameter optimisation tools to align design of CNS inhibitors with optimal ADME/PK properties. Do: [10.1021/acscchemneuro.6b00029](https://doi.org/10.1021/acscchemneuro.6b00029)

3. In the initial screen of the compound library how are the parameter cut offs set that dictate a "hit".

No strict cut-offs were used as we think this is a subjective term, however for pragmatic reasons we based parameters of a DC50 < 1uM and Dmax >50% for our analysis as a guide to capture overall success rate of finding 'hits'.

4. Can the authors comment on why only short linkers with low polarity were used?

We avoided use of excessively long linkers as our collective experience on previous PROTAC projects suggests these are not easily optimisable later on in projects. The bond lengths and linkers we chose were aimed at mirroring, broadly, the types of linkers often observed in clinical phase PROTACs as well as avoiding excessive polarity/chargeable centres as this would intuitively negatively impact cell and brain permeability.

We now modified the description of linker design as follows (line 149, top of page 4):

We designed a linker set encompassing a diverse range of properties, all featuring a secondary amine as the nucleophilic component and avoiding excessive linker length and polarity as our previous experience suggests this can cause challenges with optimisation later in a project (Fig 2B).

5. -The results from compounds with only 50-60% purity seems like it could lead to false positive or false negative results. Can you comment on the use of that >50% purity cut off point? Could this be why some wells never reached 100% DCmax? Do you think potential hits were missed simply due to their lower synthetic yields and lower reactivity of the substrates?

The level of purity sufficient to identify a hit molecule in a screen is subjective and impacted not only by the level of purity of the desired molecule but the nature of the impurities and their levels. For example, the extent to which competing by-products are present. We therefore used UV purity as a guide to evaluate trends in success of the chemistry and to identify linkers and ligands that performed well or less well to inform future modifications. For this reason we screened every well in the assay. The controls we used, most importantly the use of benzyl bromide (row E), make it highly unlikely (but not impossible) that we saw any false positives, as all negative control wells display no activity. The nature of this approach means that we cannot guarantee we did not have any individual false negatives. However, the power of the approach (and any D2B approach) is that trends and outliers are generally easy to spot. E.g. F10 immediately sticks out as it is the only completely inactive molecule in its row – but the UV purity heatmap likewise shows this underperformed in the chemistry, so interpretation would need to be qualified accordingly. Nevertheless, the main goal of identifying potent starting points/hits, is achieved.

To capture this we have amended the discussion around purity of compounds and analysis as follows:

We opted to screen all compounds/wells as there is no defined cut-offs for what constitutes sufficient purity for screening and in fact would likely rely on not only the purity of the desired compound, but the nature of the impurities. Nevertheless, as a guide, our analysis showed that a total of 147 compounds were detected by mass spectrometry, with 103 exhibiting >50% UV purity across the two plates and other groups have reported this to be a purity level at which minimal impact on assay interpretation is unlikely to be greatly affected.

6. -In addition, you mention in line 162 that you assume 100% conversion and purity when you make your 10 mM stocks, but obviously some compounds are much less pure than this. Does this not skew your resulting DC50s?

The data we have shown in figure 3 suggests that purity in this case impacted DC₅₀ but not Dmax, when comparing the crude vs pure samples. We made this assumption for practical reasons and we believe our data supports that this approach did not preclude identifying excellent hit compounds. One could alter dilution based on relative purity, but we feel this overcomplicates the workflow and also introduces other caveats (which again relate to nature not level of impurities potentially differing across compounds/wells).

7. There was no mention of compound toxicity either in cells or in vivo, which is important for assessing the utility of a new drug-like compound. In addition was any toxicity noted in the in vivo (or in cell-based) studies, especially with the high degradation of GSK3 β in the liver? Also in Fig. S2A cells are treated with CHX (cycloheximide) for up to 72h. This will presumably affect cell viability since CHX is toxic and the authors should comment on how this might affect the conclusions drawn from the data with appropriate control experiments.

In vivo there were no adverse signs observed in the 5mg/kg IV study, or the lower IV/PO study. We have also now added data (supplementary figure 3) demonstrating no impact on cell viability up to doses of 1 μ M and up to time points of 24 hours.

We appreciate the reviewers comments regarding extended time-points impacting cell viability when using CHX. For this reason we now include data (Supp fig 2) only for up to 48 hr treatment times. However, this does not affect the conclusion that GSK3 β half-lives (both WT and the hibit-tagged construct) are >24 hours.

8. The PDB ID "5K4N" was removed from the distribution of released PDB entries (status Obsolete) on 2018-12-05. Details: THE ENTRY IS OBSOLETE PER AUTHORS REQUEST."

Many thanks for noting this. This is a typo and we should have stated 5K5N. This is now corrected.

9. Missing reference for Fig S1C-D. There is some confusion on the nomenclature of protein-ligand interactions. Cation- π interactions refer to a positive charged ligand interacting with aromatic amino acid residues (e.g., Phe, Tyr, or Trp) on the target protein (e.g. 10.1038/nature07768; 10.1021/ar300265y). On the other hand, π -cation interactions, refer to an interaction between the π system of a small-molecule ligand with cationic amino acid residues (e.g., protonated Arg or Lys) in the target protein (e.g. 10.1006/jmbi.2000.4033). In addition, in Fig S1D (8DJC) the hydrogen atoms for the ligand are shown while in Fig S1C there are no hydrogen atoms. Maybe the authors could prepare both images removing hydrogen atoms for the small molecules and highlighting the π -

cation (and not as written in the text cation- π) interaction of the triazole with the R141, highlighting distances between amino group and triazole.

Thanks for alerting us to these important details. We have now added a call-out for these panels (S1C-D) on page 4. We have also altered text to reflect the correction as suggested to ' π -cation' interactions. We have also removed as requested hydrogens from the ligand in panel D. Finally, we have now also highlighted distances between R141 and triazole in panel C and R141 and the Phenyl ring (as triazole is not part of this molecules but is indicative as is attached from this position) of the ligand in panel D.

For info, we have also now added two extra panels to supporting figure 1 (E and F), with chemical structures of parent ligands shown in the co-crystal structures in panel C and D

10. Fig. S2C: GSK3 α should be GSK3 α . Also there is less degradation of GSK3 α compared to GSK3 β with compound 26, at 100nm. This behavior could be driven by the Hook effect. But if so would it be expected to be more marked with just one the two paralogues since this would translate to better selectivity for 26 with GSK3 α . However for compound 26 at 10 nM there is more degradation of GSK3 β . Thus, it would be very beneficial comment about the differences in the degradation of the two paralogues by compound 26 and link this with selectivity.

Whilst we appreciate and thank the referee for this observation, on review we do not think overall our data supports appreciable selectivity for the profiled compounds between the two paralogues and have therefore chosen to remove our comments on this. Investigating the potential for paralog specific modulators is the basis of ongoing studies.

11. Are there any data for the relative expression of the two paralogues in the brain (mRNA or protein expression)?

Based on copy number analysis from the Muqit lab (MRC-PPU), in mouse cortical neurons expression of the beta paralog is approximately double that of the alpha paralog (doi: 10.1016/j.dib.2023.109336).

12. Is anything known about the rates of synthesis of GSK3 α and GSK3 β since this can control TPD efficiency and selectivity.

Data we now shown in supporting figure 2, showing decay of GSK3 paralogues over 48 hours following treatment with CHX, suggest they are both long-lived proteins, with half lives beyond the end-point of the assay (i.e. 24 hours or longer). It is therefore unlikely that this is a major factor in PROTAC performance in this case.

13. Fig. S2I: What is the concentration of the PROTACs reported in the graph? Why is it missing traces for compounds 23 and 26? In addition, it might be worth focusing on degradation in the first hour, to have a better insight into the initial velocity

Figure 2I shows a plot of observed rates of degradation at each tested concentration vs concentration to obtain an overall rate constant for each compound. For this reason we have not adjusted the plot as requested as curves are more accurately generated and presented including all qualifying (i.e. non-hooking) data points. Apologies if this was not made clear enough, we have adjusted the legend now to reflect this. Compounds 23 and 26 were omitted as due to strong hook effects (which confound this analysis) too many concentrations had to be excluded and thus a

meaningful curve could not be obtained.

14. P.6 l.180: Is a 20-fold difference moderate? In any case it would be useful to provide an explanation for this marked difference between the two synthetic approaches.

We have now modified the text here as follows:

Observed DC_{50} values were left-shifted (2.5 to 20 fold shift) for all selected pure compounds, whilst D_{max} values remained the same between crude and pure compounds. Approximate rank ordering also remained the same amongst the set and we expect the degree of DC_{50} shift between crude and pure compounds reflects the actual vs assumed (10 mM) concentration of each compound in the crude stock solution produced for testing, as well as any remaining competing impurities.

15. Fig 3A and C are not very clear since so many data are included in the graphs. However : Fig 3B and D are much clearer. It is confusing that your panels are not in order.

This has now been adjusted to improve readability and also in line with referee 1 comments.

16. Suggest that all instances of “in vivo” and “in vitro” are italicized

As we understand, the journal requests non-italicised fonts for in vivo and in vitro, but will correct as guided by editorial staff.

17. There may be a typo in line 54 and 144.

We could not identify any typos on these lines, but are of course keen to modify if specific examples can be provided.

18. In Figure 5, there is * above some panels, but it is not specified what p value this represents.

Thanks for alerting to this, we have now added p values to this figure panel (now figure 6).

19. A section is included in your methods for tissue processing for the PD study, but you don't include any details of how blood was worked up for MS analysis. Was whole blood analyzed MS without any processing?

We have now added a section in the methods describing MS based blood analysis and sample preparation steps taken.

20. Suggest combining Table S5 and S6 for simplicity. Is there a way to also combine Tables S2-S4?

We have combined S3 and S4, to now give a new Table S3 and also combined S5 and S6 to give a new Table S4.

21. It would be beneficial to discuss the serum shift assay in more detail and the importance of the results derived from that assay.

This assay is highly useful when facing difficulties with very highly bound PROTACs, though on this occasion we found good agreement between the serum shift assay and the equilibrium dialysis data. As stated, we have described this assay previously and cited this, but have made a minor modification to the text to explain the utility of the data generated:

In reasonable agreement with this, the serum shift degradation assay showed a shift in DC_{50} of 8.2x with 10% Mouse serum (Fig 6B), therefore predicting a shift of ~80x in whole blood and equating to a free fraction of ~1.2% in mouse plasma, giving us extra confidence for predicting free exposure in vivo to guide our dosing regimen.

22. In your chemistry SI, “calculated” is consistently and incorrectly spelled as “calcuLated”.

Thanks for spotting this, now corrected.

23. Chemical formulas should have the numbers subscripted.

These have now been corrected, thanks.

24. It would be beneficial to include schemes in the chemistry SI.

Many thanks, we have now added schemes and corrected according to feedback from comment 22 and 23.

Reviewer #5 (Remarks to the Author):

This manuscript presents a novel screening strategy for identifying novel E3 ligase-target protein modalities for targeted protein degradation, simultaneously varying linker, E3, and target recruiting ligands. The authors describe development and validation of a CNS-penetrant GSK3 degrader using this orthogonally reactive linker platform for screening. The method provides a framework to test design principles for PROTAC generation via sequential synthesis of bifunctional molecules, thereby diverging from the concept of E3 ligase-linker pre-conjugation and identifying alternative chemical modalities mediating GSK3 attenuation compared to previously published GSK3 degraders. The authors identify potent degrader (compound 24) with sub-nM DC_{50} , rapid GSK3 degradation kinetics in cells, and bioactivity in mice. Mechanism of action analyses reveal dependence upon GSK3 kinase domain and CRBN for degradation. Deeper insight into proteome-level changes (in addition to GSK3 β attenuation) in mouse liver/brain is required to validate the conclusion that KH1 is selective and tolerated at the presented dose in vivo.

Many thanks to the referee for the overall supportive and insightful comments. Seeking to address their major area of feedback we have now included additional proteomic and phosphoproteomic profiling from both cellular and in vivo studies, as well as additional cell viability assay data. In addition to no adverse events being observed in either the 5mg/kg i.v. dosed animal or the lower dose PK studies, we believe the data presented fully supports excellent selectivity and tolerability of KH1 at the doses presented. We detail this with respect to each point below and hope this new data addresses the referees main queries.

Major comments:

1.) A broader proteomic analysis would provide stronger validation for selectivity of compound 24 as

a chemical probe and elucidate possible off target effects with longer durations of treatment. While the selectivity for compound 24 is strong (figure 4D), this compound is modestly perturbing the proteome at 2h. What happens to the proteome at later timepoints including 4h (as presented in MEFs and mouse treatments), or 24h as presented in SH-SY5Y experiments? There are several proteins upregulated with treatment of compound 24; while only 1.5-fold, this effect may be more pronounced with longer durations of treatment.

We appreciate the referee encouraging us to investigate both whole and phosphoproteomes in more depth and have been excited to follow this up as now shown. To address the key queries in points 1 and 3 we have now provided data on whole proteomes and phosphoproteomes following KH1 (24) and negative control (27) treatment at 2, 4 and 24 hours in HEK cells. This was performed as an extension to the data we previously included evaluating whole proteome changes in HEK cells at 2 hours as acknowledged by the reviewer. We used a dose of 10 nM for both compounds as these concentrations are around/above the DC_{90} and initially at a timepoint (2 hrs) at which near maximal degradation is observed. This is a gold standard and widely accepted approach for assessing direct degradation of on vs off targets in the PROTAC field. We were pleased to observe that KH1 retains excellent proteome wide specificity up to 24 hours, with only a small expansion in the number of significantly modulated proteins. Given this represents a time point at least 22 hours post GSK3 removal we cannot ascertain whether proteins which show small changes in abundance are direct or indirect (i.e, altered as a downstream consequence of GSK3 removal) target proteins. We did observe small increases in beta catenin at later timepoints and have commented on this in the main text. (please see fig5 and supporting fig4).

In addition, for each of these treatments we have enriched phosphopeptides and analysed impact on the phosphoproteomes at each time-point. We thank the referee for this suggestion as we feel it has enabled us to demonstrate an important utility of the probe, that is, as a way to kinetically illuminate impact of GSK3 shutdown. We hope our data exemplifies what could be learned from related and more in depth experiments in more complex models.

In summary, we greatly appreciate the encouragement to dive deeper here and believe this new data has enabled us to strengthen the utility of our probe molecule for investigating GSK3 biology in a manner that would be challenging/impossible via inhibition or genetic intervention. We believe this data supports both the specificity of KH1 at longer treatments times in cells as well as it's ability to modulate GSK3 substrate phosphorylation following degradation, highlighting the dynamic and differential susceptibility of different substrates to GSK3 post-translational removal.

Unique peptides quantified for GSK3 α vs. GSK3 β ? Presenting the peptide info used for quant would be useful in a supplemental figure.

We have now created a separate table (See relevant tabs in Raw data file) for all unique identified peptides identified in proteomics studies.

2.) Does compound 24 impact GSK3 activity prior to its degradation? Chemical inhibition has been shown to increase inhibitory phos. at ser 9. Is auto-phosphorylation responsible for the rapid degradation kinetics (GSK3 inhibitor treatment in figure 4F/G), or does GSK3 inhibitor binding prevent compound 24 from accessing the kinase domain? The figure legend notes GSKi 47, methods state compound 20-they are synonymous but carry consistent nomenclature.

'Compound 47' is the name given to the parent GSK3 inhibitor structure by previous authors and we show the structure of this in figure S1F. Compound 20 (as named in this manuscript) is an azide

substituted analogue of this parent inhibitor which we designed, synthesised and used for synthesis of our library as shown in figure 2F, with the structure of compound 20 shown in figure 2D. Compound 29 is the bromide substituted analogue of compound 20. Compound 29 is also used in Figure 4F and G. We have updated compound numbers accordingly.

To address the reviewers question, we have now added data (figure 4G) showing a comparison of effects on phosphorylation of Ser9 and Tyr216 of GSK3 β following treatment with KH1 (24), negative control (27) and GSK3 inhibitor compound 29, at 30 minutes and 2 hours pre-treatment. It should be noted that Serine 9 is an inhibitory site regulated by other kinases, whereas Tyr 216 is an activation site that is regulated by autophosphorylation. Our data shows no impact on either site after 30 minutes treatment, with only concurrent reduction in phospho and total GSK3 levels as PROTAC mediated degradation ensues. This is supported by comparison with non-degrading negative control 27 and GSK3 inhibitor treatments. We believe this supports that GSK3 inhibitor/PROTAC co-treatment prevents GSK3 degradation via the inhibitor blocking PROTAC engagement and that no appreciable inhibition of GSK3 catalytic activity takes place prior to KH1 mediated degradation, likely due to very low occupancy levels achieved at 10 nM (which is still sufficient for complete degradation).

A phospho-proteome analysis of the differentiated SH-SY5Y cells (western shown in Fig 4i or in HEK293 cells if more permissible) with corresponding whole proteome would strengthen the conclusion that GSK3 substrates are largely de-phosphorylated (or stabilized in the case of those that are degraded following phosphorylation) following treatment.

A short timepoint (prior to robust GSK3 degradation) and both 2-4h/24h could be informative to assess phosphorylation-dependent changes and global protein stability following GSK3 attenuation.

Please see our response to point 1 which also captures our newly provided phosphoproteomics data.

3.) Compound 24 (KH1) leads to significant GSK3 β attenuation in mouse liver and moderate reduction in mouse brain 4 hours post treatment. If no additional dose is given, do the mice survive normally? It would be interesting to see a whole proteome analysis for naïve, 4h treated, and 4h treated followed by several hours of recovery to examine how the liver and brain proteome may be remodeled in vivo. This may highlight potential off-target or stress signatures before additional doses/treatment regimens are examined in vivo.

We greatly appreciate the interest and relevance of understanding the impact, safety and potential of repeated or longer term treatment in vivo using KH1. To try and further support and address the reviewers question we have now also provided whole and phosphoproteome analysis of the liver samples from the presented PD study (please see new figure 6), which we had retained and had sufficient material to conduct. This data matches the overall observations from the cell studies, in that the molecule is highly specific across the proteome and that GSK3 β is by far the most modified protein. In addition, we have also now provided cell viability data at doses and timepoints well above and beyond those achieved/used in vivo. Also, we noted no adverse signs in the 5mg/kg IV study, or the lower IV/PO study and overall, we feel the additional data provided provides no obvious safety concerns. We have limited our claims to the data and dosing regimens exemplified specifically in our data set and have not commented or claimed use beyond those. We do, however, fully acknowledge that it would be interesting and valuable to understand what happens with repeat doses and/or extended treatment times in vivo, but we feel this would move away from our main claims and form the basis of a new study which is beyond our current funding.

Minor comments:

1.) Comparison of ligand 19 and 20—what is the reason for the increased frequency of potent compounds with the latter? It is apparent from the results, but a line describing why this may be the case would be beneficial to the reader. Supplemental Figure S1C, D are not discussed in the manuscript. The results section reads “For GSK3 targeting ligands, we designed azide containing building blocks based on PF-367 and CMP-47(Fig 2D)” —reference the supplemental data here to demonstrate engagement of the GSK3 kinase domain and clarify for the reader that these are ligands 19, 20 respectively.

We agree that there appears to be a bias towards increased hit rate using ligand 20. We speculate this may be reflective of increased GSK3 engagement with this series, but future studies will be aimed at answering such questions.

Many thanks for spotting omission of call-out of figures S1C and D, we have now corrected this in the context of our discussion regarding ligand and exit vector choice and design. We hope this makes it clear to the reader that previously published co-crystal structures demonstrate engagement and mode of binding on which our designs were based.

2.) GSK3 β expression as detected by endogenous antibody is lower in HiBiT tagged cells vs. WT. Are both alleles targeted? Stability following CHX is comparable b/w tagged and endo GSK3 β in the 72h experiment (S2b) but baseline expression is lower. Showing the 72h experiment should suffice given GSK3 β is a long-lived protein.

MiSeq data is now provided identifying that 2/3 of the genome has been modified with hibit insertion to c-terminal of GSK3 β (note, HEK293 cells are triploid). Due to potential confounding toxicity of CHX on cells at longer time-points we have included now up to 48 hours treatments, which still demonstrate unaltered half-lives for hibit vs WT GSK3 β . We do not know exactly why we observe slightly lower GSK3 β in the hibit-in cell line. However, as this was a cell line produced for screening and both detection and an unchanged half-life lended itself suitable for this use we do not view this as a considerable issue. Importantly, we have validated our hits against wild type endogenous GSK3 β and GSK3 α in multiple cell lines and observed similar potencies to support that this screening cell line performed as designed to identify hits.

3.) Figure S3: panel F should be moved to the left and panels H, I shifted to the right. The figure is hard to follow given the order of panels. Furthermore, labels on panel F are likely reversed—compound 24 should exhibit GSK3 degradation while compound 27 is the control, non-degradative compound. Presenting cell viability data in response to 24h of 10 or 100nM compound 24 treatment would be helpful to understand potential toxicity with longer duration or increased concentration.

Many thanks indeed for spotting this error - we have modified what is now supplementary figure 4 as proposed.

We have also now added cell viability data as requesting using 10 nM, 100 nM and in addition we have chosen to add data at 1 μ M also. We used time points of 2, 4 and 24 hours. This data is presented in supplementary figure S3. This data shows no impact on cell viability in any of the treatments.

4.) Figure legends describing statistical data could be expanded slightly to explain error bars and significance notations.

Thanks for noting this. We have endeavoured to now ensure sufficient details as including a Statistical Analysis section prior to the references. We hope this sufficiently addresses the reviewer's request.